# Lower Bounds on Adaptive Sensing for Matrix Recovery

**Praneeth Kacham**
Computer Science Department
Carnegie Mellon University
pkacham@cs.cmu.edu

**David P. Woodruff**
Computer Science Department
Carngie Mellon University
dwoodruf@cs.cmu.edu

## Abstract

We study lower bounds on adaptive sensing algorithms for recovering low rank matrices using linear measurements. Given an $n \times n$ matrix $A$, a general linear measurement $S(A)$, for an $n \times n$ matrix $S$, is just the inner product of $S$ and $A$, each treated as $n^2$-dimensional vectors. By performing as few linear measurements as possible on a rank-$r$ matrix $A$, we hope to construct a matrix $\hat{A}$ that satisfies

$$\|A - \hat{A}\|_{\mathsf{F}}^2 \le c\|A\|_{\mathsf{F}}^2, \tag{1}$$

for a small constant $c$. Here $\|A\|_{\mathsf{F}}$ denotes the Frobenius norm $(\sum_{i,j} A_{i,j}^2)^{1/2}$. It is commonly assumed that when measuring $A$ with $S$, the response is corrupted with an independent Gaussian random variable of mean 0 and variance $\sigma^2$. Candés and Plan (IEEE Trans. Inform. Theory 2011) study non-adaptive algorithms for low rank matrix recovery using random linear measurements. They use the restricted isometry property (RIP) of Random Gaussian Matrices to give tractable algorithms to estimate $A$ from the measurements.

At the edge of the noise level where recovery is information-theoretically feasible, it is known that their non-adaptive algorithms need to perform $\Omega(n^2)$ measurements, which amounts to reading the entire matrix. An important question is whether adaptivity helps in decreasing the overall number of measurements. While for the related problem of sparse recovery, adaptive algorithms have been extensively studied, as far as we are aware adaptive algorithms and lower bounds on them seem largely unexplored for matrix recovery. We show that any adaptive algorithm that uses $k$ linear measurements in each round and outputs an approximation as in (1) with probability $\ge 9/10$ must run for $t = \Omega(\log(n^2/k)/\log\log n)$ rounds. Our lower bound shows that *any* adaptive algorithm which uses $n^{2-\beta}$ (for any constant $\beta > 0$) linear measurements in each round must run for $\Omega(\log n/\log\log n)$ rounds to compute a good reconstruction with probability $\ge 9/10$. Hence any adaptive algorithm that has $o(\log n/\log\log n)$ rounds must use an overall $\Omega(n^2)$ linear measurements. Our techniques also readily extend to obtain lower bounds on adaptive algorithms for tensor recovery.

Our hard distribution also allows us to give a measurement-vs-rounds trade-off for many sensing problems in numerical linear algebra, such as spectral norm low rank approximation, Frobenius norm low rank approximation, singular vector approximation, and more.

## 1 Introduction

Sparse recovery, also known as compressed sensing, is the study of under-determined systems of equations subject to a sparsity constraint. Suppose we know that an unknown vector $x \in \mathbb{R}^n$

37th Conference on Neural Information Processing Systems (NeurIPS 2023).

has at most $r \ll n$ non-zero entries. We would like to use a measurement matrix $M \in \mathbb{R}^{k \times n}$ to recover the vector $x$ given measurements $y = Mx$. The number $k$ of measurements is an important parameter we would like to optimize as it models the equipment cost in physical settings and running times in computational settings. Ideally one would like for the number $k$ of measurements to scale proportionally to the sparsity of the unknown vector $x$.

One way of modeling sparse recovery is as "stable sparse recovery". Here we want a distribution $\mathcal{M}$ over $k \times n$ matrices such that for any $x \in \mathbb{R}^n$, with probability $\geq 1 - \delta$ over $\boldsymbol{M} \sim \mathcal{M}$, we can construct a vector $\hat{x}$ from $\boldsymbol{M}x$ such that

$$\|x - \hat{x}\|_p \leq (1 + \varepsilon) \min_{r\text{-sparse } x'} \|x - x'\|_p.$$

Note that the above formulation is more robust as it does not require the underlying vector $x$ to be exactly $r$-sparse and instead asks to recover the top $r$ coordinates of $x$.

For $p = 2$, it is known that $k = \Theta(\varepsilon^{-1} r \log(n/r))$ measurements is both necessary and sufficient [6, 9, 22, 4]. See [23] and references therein for upper and lower bounds for other values of $p$.

In the same vein, the problem of low rank matrix recovery has been studied. See [5, 35] and references therein for earlier work and numerous applications. In this problem, the aim is to recover a *low rank* matrix $A$ using linear measurements. Here we want a distribution $\mathcal{M}$ over linear operators $M : \mathbb{R}^{n \times n} \to \mathbb{R}^k$, such that for any *low* rank matrix $A$, with probability $\geq 1 - \delta$ over $\boldsymbol{M} \sim \mathcal{M}$, we can construct a matrix $\hat{A}$ from $\mathcal{M}(A)$ such that the reconstruction error $\|A - \hat{A}\|_{\mathsf{F}}^2$ is small. Note that a linear operator $M : \mathbb{R}^{n \times n} \to \mathbb{R}^k$ can be equivalently represented as a matrix $M \in \mathbb{R}^{k \times n^2}$ such that $M \cdot \text{vec}(A) = M(A)$ where $\text{vec}(A)$ denotes the appropriate flattening of the matrix $A$ into a vector. We call the rows of $M$ *linear measurements*. Without loss of generality, we can assume that the rows of the matrix $M$ are orthonormal, as the responses for non-orthonormal queries can be obtained via a simple change of basis.

In addition, to model the measurement error that occurs in practice, it is a standard assumption that when querying with $M$, we receive $M \cdot \text{vec}(A) + \boldsymbol{g}$, where $\boldsymbol{g}$ is a $k$-dimensional vector with independent Gaussian random variables of mean 0 and variance $\sigma^2$, and we hope to reconstruct $A$ with small error from $M \cdot \text{vec}(A) + \boldsymbol{g}$. Clearly, when $A$ has rank $r$, we need to perform $\Omega(nr)$ linear measurements, as the matrix $A$ has $\Theta(nr)$ independent parameters. Hence, the aim is to perform not many more linear measurements than $nr$ while being able to obtain an estimate $\hat{A}$ for $A$ with low estimation error.

Given a rank-$r$ matrix $A$, [5] show that if the $k \times n^2$ matrix $\boldsymbol{M}$ is constructed with independent standard Gaussian entries, then with probability $\geq 1 - \exp(-cn)$, an estimate $\hat{A}$ can be constructed from $\boldsymbol{M} \cdot \text{vec}(A) + \boldsymbol{g}$ such that $\|A - \hat{A}\|_{\mathsf{F}}^2 \leq O(nr\sigma^2/k)$. They use the restricted isometry property (RIP) of the Gaussian matrix $\boldsymbol{M}$ to obtain algorithms that give an estimate $\hat{A}$. The error bound is intuitive since the reconstruction error increases with increasing noise and proportionally goes down when the number of measurements $k$ increases.

While we formulated the sparse recovery and matrix recovery problems in a non-adaptive way, there have been works which study adaptive algorithms for sparse recovery. Here we can produce matrices $\boldsymbol{M}^{(i)}$ adaptively based on the responses received $\boldsymbol{M}^{(1)}x, \boldsymbol{M}^{(2)}x, \ldots, \boldsymbol{M}^{(i-1)}x$ and the hope is that allowing for adaptive algorithms with a small number of adaptive rounds, we obtain algorithms that overall perform fewer linear measurements than non-adaptive algorithms with the same reconstruction error. It is additionally assumed that the noise across different rounds is independent. For sparse recovery, in the case of $p = 2$, it is known that over $O(\log \log n)$ rounds adaptivity, a total of $O(\varepsilon^{-1} r \log \log(n \log(n/r)))$ linear measurements suffices [14, 21], improving over the requirement of $\Theta(\varepsilon^{-1} r \log(n/r))$ linear measurements for non-adaptive algorithms. For a constant $\varepsilon$, for any number of rounds, it is known that $\Omega(r + \log \log n)$ linear measurements are required [23]. Recently, [16] gave a lower bound on the total number of measurements if the number of adaptive rounds is constant.

While there has been a lot of interest in adaptive sparse recovery, both from the algorithms and the lower bounds perspective, the adaptive matrix recovery problem surprisingly does not seem to have any lower bounds. In adaptive matrix recovery, similar to adaptive sparse recovery, one is allowed to query a matrix $M^{(i)}$ in round $i$ based on the responses received in the previous rounds, and again the hope is that with more rounds of adaptivity, the total number of linear measurements that is to

be performed decreases. There is some work that studies adaptive matrix recovery with 2 rounds of adaptivity (see [34] and references therein) but the full landscape of adaptive algorithms for matrix recovery seems unexplored.

We address this from the lower bounds side in this work. We show lower bounds on adaptive algorithms that recover a rank-$r$ matrix of the form $(\alpha/\sqrt{n}) \sum_{i=1}^{r} \boldsymbol{u}_i \boldsymbol{v}_i^{\mathsf{T}}$, where the coordinates of $\boldsymbol{u}_i$ and $\boldsymbol{v}_i$ are sampled independently from the standard Gaussian distribution. Without loss of generality[1], we assume that the measurement matrix $M^{(i)}$ in each round $i$ has orthonormal rows. We also assume that for $i \neq j$, the measurement matrices $M^{(i)}(M^{(j)})^{\mathsf{T}} = 0$, i.e., measurements across adaptive rounds are orthonormal, since non-orthonormal measurements can be reconstructed from orthonormal measurement matrices by a change of basis.

We now give an alternate way of looking at the adaptive sparse recovery problem: Fix a set of vectors $u_1, \ldots, u_r$ and $v_1, \ldots, v_r$ and let the underlying matrix we want to reconstruct be $(\alpha/\sqrt{n}) \sum_{i=1}^{r} u_i v_i^{\mathsf{T}}$ for a large enough constant $\alpha$. In round 1, we query a matrix $\boldsymbol{M}^{(1)} \in \mathbb{R}^{k \times n^2}$ drawn from an appropriate distribution and receive the response vector $\boldsymbol{r}^{(1)} = \boldsymbol{M}^{(1)} \operatorname{vec}((\alpha/\sqrt{n}) \sum_{i=1}^{r} u_i v_i^{\mathsf{T}}) + \boldsymbol{g}^{(1)}$ where $\boldsymbol{g}^{(1)}$ is a vector of independent Gaussian random variables with mean 0 and variance $\sigma^2$. In round 2, as a function of $\boldsymbol{M}^{(1)}$ and $\boldsymbol{r}^{(1)}$ and our randomness, we query a random matrix $\boldsymbol{M}^{(2)}[\boldsymbol{r}^{(1)}] \in \mathbb{R}^{k \times n^2}$ and receive a response vector $\boldsymbol{r}^{(2)} = (\boldsymbol{M}^{(2)}[\boldsymbol{r}^{(1)}]) \operatorname{vec}((\alpha/\sqrt{n}) \sum_{i=1}^{r} u_i v_i^{\mathsf{T}}) + \boldsymbol{g}^{(2)}$ where $\boldsymbol{g}^{(2)}$ is again a vector with independent Gaussian entries and independent of $\boldsymbol{g}^{(1)}$, and so on. Crucially, using the rotational invariance of Gaussian random vectors, if $\boldsymbol{G}$ is an $n \times n$ matrix with independent Gaussian random variables with mean 0 and variance $\sigma^2$, the response $\boldsymbol{r}^{(1)}$ has the same distribution as $(\boldsymbol{M}^{(1)}) \cdot \operatorname{vec}((\alpha/\sqrt{n}) \sum_{i=1}^{r} u_i v_i^{\mathsf{T}} + \boldsymbol{G})$ and as $\boldsymbol{M}^{(2)}[\boldsymbol{r}^{(1)}]$ is chosen to be orthonormal to $\boldsymbol{M}^{(1)}$, the distribution of $\boldsymbol{r}^{(2)}$ conditioned on $\boldsymbol{r}^{(1)}$ is the same as that of $(\boldsymbol{M}^{(2)}[\boldsymbol{r}^{(1)}]) \cdot (\operatorname{vec}((\alpha/\sqrt{n}) \sum_{i=1}^{r} u_i v_i^{\mathsf{T}} + \boldsymbol{G}))$.

Thus, any adaptive matrix recovery algorithm can be seen as performing *non-noisy* adaptive measurements on the matrix $(\alpha/\sqrt{n}) \sum_{i=1}^{r} u_i v_i^{\mathsf{T}} + \boldsymbol{G}$ where the Gaussian matrix $\boldsymbol{G}$ is sampled independently of the measurement algorithm. From the responses the algorithm receives, it then tries to reconstruct the matrix $(\alpha/\sqrt{n}) \sum_{i=1}^{r} u_i v_i^{\mathsf{T}}$. This way of looking at adaptive sparse recovery immediately yields an adaptive algorithm: when the smallest singular value of $(\alpha/\sqrt{n}) \sum_{i=1}^{r} u_i v_i^{\mathsf{T}}$ is a constant times larger than $\|\boldsymbol{G}\|_2$, then the power method with a block size of $r$ outputs an approximation of $(\alpha/\sqrt{n}) \sum_{i=1}^{r} u_i v_i^{\mathsf{T}}$ in $O(\log n)$ rounds. Note that $r$ matrix-vector products can be implemented using $nr$ linear measurements. More generally, Gu [12] showed that using power iteration with a block size of $k/n$ (for $k \geq nr$), we can obtain an approximation in $O(\log(n^2/k))$ adaptive rounds. Thus the already existing algorithms exhibit a no. of measurements vs no. of rounds trade-off.

From results in random matrix theory, we have that $\|\boldsymbol{G}\|_2 \approx \sigma \sqrt{n}$ with high probability. And as we are interested in reconstruction when the vectors $u_1, \ldots, u_r$ and $v_1, \ldots, v_r$ follow the Gaussian distribution we also have that $\|u_i\|_2 \approx \|v_i\|_2 \approx \sqrt{n}$ simultaneously with large probability. Thus, to make the extraction of $(\alpha/\sqrt{n}) \sum_{i=1}^{r} u_i v_i^{\mathsf{T}}$ information-theoretically possible, we need to assume that $\alpha \gtrsim \sigma$. Hence, in this work we assume that $\sigma = 1$ and that $\alpha$ is a large enough constant, and we study lower bounds on adaptive matrix recovery algorithms.

If the vectors $u_1, \ldots, u_r$ and $v_1, \ldots, v_r$ are sampled from the standard $n$ dimensional Gaussian distribution and $r \leq n/2$, we also have that with high probability $\|(\alpha/\sqrt{n}) \sum_{i=1}^{r} u_i v_i^{\mathsf{T}}\|_F^2 \approx \alpha^2 nr$. Now the algorithms of [5], which use a uniform random Gaussian matrix $\boldsymbol{M}$ with $m$ rows as the measurement matrix, for $(\alpha/\sqrt{n}) \sum_{i=1}^{r} u_i v_i^{\mathsf{T}}$ reconstruct a matrix $\hat{A}$ such that $\|\hat{A} - (\alpha/\sqrt{n}) \sum_{i=1}^{r} u_i v_i^{\mathsf{T}}\|_F^2 \leq C \frac{nr\sigma_m^2}{m}$ where $\sigma_m$ is the measurement error. As we assumed above that $\sigma = 1$ when measuring with a matrix with orthonormal rows, we assume that the measurement error $\sigma_m$ when measuring with a Gaussian matrix is $n$, as each row of $\boldsymbol{M}$ has $n^2$ independent Gaussian coordinates and therefore has a norm $\approx n$. Thus, using reconstruction algorithms from [5], we obtain a matrix $\hat{A}$ satisfying $\|\hat{A} - (\alpha/\sqrt{n}) \sum_{i=1}^{r} u_i v_i^{\mathsf{T}}\|_F^2 \leq C \frac{n^3 r}{m}$. Now, to make $\|\hat{A} - (\alpha/\sqrt{n}) \sum_{i=1}^{r} u_i v_i^{\mathsf{T}}\|_F^2 \leq (1/10)\|(\alpha/\sqrt{n}) \sum_{i=1}^{r} u_i v_i^{\mathsf{T}}\|_F^2$, we need to set $m = \Theta(n^2/\alpha^2)$. Hence, in the parameter regimes we

---

[1]In general, in the sparse recovery and matrix recovery models, the algorithms may make the same query and obtain responses with independent noise added which can be used to say obtain more accurate result using the median/mean estimator. But all the algorithms we are aware of for sparse recovery and matrix recovery do not explore independence of noise across queries in this way.

study, the algorithms of [5] need to perform $\Omega(n^2)$ non-adaptive queries, which essentially means that they have to read a constant fraction of the $n^2$ entries in the matrix. While the power method performs $O(nr)$ linear measurements in each round over $O(\log n)$ adaptive rounds to output an approximation $\hat{A}$. The question we ask is:

"Is the power method optimal? Are there algorithms that have $o(\log n)$ adaptive rounds and use $n^{2-\beta}$ measurements in each round?"

We answer this question by showing that any algorithm that has $o(\log n / \log \log n)$ adaptive rounds must use $\geq n^{2-\beta}$ linear measurements, for any constant $\beta > 0$, in each round. The lower bound shows that power method is essentially optimal if we want to use $n^{2-\beta}$ linear measurements in each round and that any algorithm with $o(\log n / \log \log n)$ adaptive rounds essentially reads the whole matrix.

We further obtain a rounds vs measurements trade-off for many numerical linear algebra problems in the sensing model. In this model, there is an $n \times n$ matrix $A$ with which we can interact only using general linear measurements and we want to solve problems such as spectral norm low rank approximation of $A$, Frobenius norm low rank approximation of $A$, etc. In general, numerical linear algebra algorithms assume that they either have access to the entire matrix or that the matrix is accessible in the matrix-vector product model where one can query a vector $v$ and obtain the result $A \cdot v$. Recently, the vector-matrix-vector product model has received significant attention as well. Linear measurements are more powerful than both the matrix-vector product model and the vector-matrix-vector product model. Any matrix vector product $A \cdot v$ can be computed using $n$ linear measurements of $A$ and any vector-matrix-vector product $u^\mathsf{T} A v$ can be computed using a single linear measurement of $A$. Thus, the model of general linear measurements may lead to faster algorithms.

For certain problems in numerical linear algebra, general linear measurements are significantly more powerful than the matrix-vector product model. Indeed, for computing the trace of an $n \times n$ matrix $A$, one can do this exactly with a single deterministic general linear measurement, just by adding up the diagonal entries of $A$. However, in the matrix-vector product model, it is known that $\Omega(n)$ matrix-vector products are needed to compute the trace exactly [29], even if one uses randomization. A number of problems were studied in the vector-matrix-vector product model in [24], and in [32] it was shown that to approximate the trace of $A$ up to a $(1 + \varepsilon)$-factor with probability $1 - \delta$, one needs $\Omega(\varepsilon^{-2} \log(1/\delta))$ queries. This contrasts sharply with the single deterministic general linear measurement for computing the trace exactly. Thus, there are good reasons to conjecture that general linear measurements may lead to algorithms requiring fewer rounds of adaptivity compared to algorithms in the matrix-vector product query model. Surprisingly, our lower bounds show that for many numerical linear algebra problems, linear measurements do not give much advantage over matrix-vector products.

## 1.1 Our Results

We assume that there is an unknown rank-$r$ matrix $A \in \mathbb{R}^{n \times n}$ to be recovered. Given any linear measurement $q \in \mathbb{R}^{n^2}$, we receive a response $\langle q, \mathrm{vec}(A) \rangle + \boldsymbol{g}$, where $\boldsymbol{g} \sim N(0, \|q\|_2^2)$. We further assume that the noise for two different measurements is independent. Without loss of generality, we also assume throughout that all the queries an algorithm makes across all rounds form a matrix with orthonormal rows. Our main result for adaptive matrix sensing is as follows:

**Theorem 1.1.** *There exists a constant $c$ such that any randomized algorithm which makes $k \geq nr$ noisy linear measurements of an arbitrary rank-$r$ matrix $A$ with $\|A\|_\mathsf{F}^2 = \Theta(nr)$ in each of $t$ rounds, and outputs an estimate $\hat{A}$ satisfying $\|A - \hat{A}\|_\mathsf{F}^2 \leq c\|A\|_\mathsf{F}^2$ with probability $\geq 9/10$ over the randomness of the algorithm and the Gaussian noise, requires $t = \Omega(\log(n^2/k)/(\log \log n))$.*

**Dependence on noise**  In our results, we assumed that given a linear measurement $q \in \mathbb{R}^{n^2}$, the response is distributed as $N(\langle q, \mathrm{vec}(A) \rangle, \|q\|_2^2)$. Our lower bounds also hold when the response is distributed as $N(\langle q, \mathrm{vec}(A) \rangle, \sigma^2 \|q\|_2^2)$ for any parameter $\sigma$ such that $c' \leq \sigma \leq 1$, where $c'$ is a constant. This can be seen by simply scaling the matrix $A$ in the theorem above and adjusting the constants while proving the theorem.

| Application | Failure Probability | Lower Bound |
|---|---|---|
| 2-approximate spectral LRA | 0.1 | $c \log(n^2/k)/\log\log n$ |
| 2-approximate spectral LRA | $1/\mathrm{poly}(n)$ | $c \log(n^2/k)$ |
| 2-approximate Schatten $p$ LRA | 0.1 | $\frac{c \log(n^2/k)}{(1/p)\log(n)+\log\log n}$ |
| 2-approximate Ky-Fan $p$ LRA | 0.1 | $\frac{c \log(n^2/k)}{\log(p)+\log\log n}$ |
| $1 + 1/n$-approximate Frobenius LRA | 0.1 | $c \log(n^2/k)/\log\log n$ |
| 0.1-approximate $i$-th singular vectors | 0.1 | $c \log(n^2/k)/\log\log n$ |
| 2-approximate spectral reduced rank regression | 0.1 | $c \log(n^2/k)/\log\log n$ |

Table 1: Number of rounds lower bound for algorithms using $k$ general linear measurements in each round. Our bound for 2-approximate spectral LRA algorithm with constant failure probability is optimal up to a $\log\log n$ factor, and our 2-approximate spectral low rank approximation (LRA) lower bound for algorithms with failure probability $1/\mathrm{poly}(n)$ is optimal up to a constant factor.

**Tensor Recovery** The problem of tensor recovery with linear measurements has also been studied (see [25, 11] and references therein) where given a low rank tensor, the task is to recover an approximation to the tensor with few linear measurements. Our techniques can also be used to obtain lower bounds on adaptive algorithms for tensor recovery. Our main tool, Lemma 3.1, readily extends to tensors of higher orders by using the corresponding tail bounds from [31].

**Numerical Linear Algebra** We also derive lower bounds for many numerical linear algebra problems in the linear measurements model. Table 1 shows our lower bounds on the number of adaptive rounds required for any randomized algorithm using $k$ general linear measurements in each round. See Appendix E for precise statements and proofs.

### 1.2 Notation

Throughout the paper, we use uppercase letters such as $G, M, Q$ to denote matrices and lowercase letters such as $u, v$ to denote vectors. For vectors $u, v \in \mathbb{R}^n$, $u \otimes v \in \mathbb{R}^{n^2}$ denotes the tensor product of $u$ and $v$. For an arbitrary matrix $M \in \mathbb{R}^{m \times n}$, the vector $\mathrm{vec}(M) \in \mathbb{R}^{mn}$ denotes a flattening of the matrix $M$ with the convention that $\mathrm{vec}(uv^\mathsf{T}) = u \otimes v$. We use boldface symbols such as $\boldsymbol{M}, \boldsymbol{G}, \boldsymbol{u}, \boldsymbol{v}$ to denote that these objects are explicitly sampled from an appropriate distribution. We use $\boldsymbol{g}_k$ to denote a multivariate Gaussian in $k$ dimensions where each coordinate is independently sampled from $N(0, 1)$. We also use $G^{(j)}$ to denote a collection of $j$ independent multivariate Gaussian random variables of appropriate dimensions.

### 1.3 Our Techniques

Using the rotational invariance of the Gaussian distribution, we argued that any adaptive randomized low rank matrix recovery algorithm with access to a hidden matrix $(\alpha/\sqrt{n}) \sum_{i=1}^{r} u_i v_i^\mathsf{T}$ using noisy linear measurements can be seen as a randomized algorithm that has access to a random matrix $(\alpha/\sqrt{n}) \sum_{i=1}^{r} u_i v_i^\mathsf{T} + \boldsymbol{G}$ using *perfect* linear measurements where each coordinate of $\boldsymbol{G}$ is independently sampled from a Gaussian distribution.

Let $\mathcal{A}$ be any algorithm that satisfies the matrix recovery guarantees with, say, a success probability $\geq 9/10$. Let $\mathcal{A}(X, \boldsymbol{\sigma}, \boldsymbol{\gamma})$ be the output of the randomized algorithm $\mathcal{A}$, where the hidden matrix is $X$, the random seed of $\mathcal{A}$ is denoted by $\boldsymbol{\sigma}$, and $\boldsymbol{\gamma}$ denotes the measurement randomness. We have that if $X$ has rank $r$ and satisfies $\|X\|_\mathsf{F}^2 = \Theta(nr)$, then

$$\Pr_{\boldsymbol{\sigma}, \boldsymbol{\gamma}}[\mathcal{A}(X, \boldsymbol{\sigma}, \boldsymbol{\gamma}) \text{ is correct}] \geq 9/10.$$

We say $\mathcal{A}(X, \boldsymbol{\sigma}, \boldsymbol{\gamma})$ is correct when the output $\hat{X}$ satisfies $\|\hat{X} - X\|_\mathsf{F}^2 \leq (1/100)\|X\|_\mathsf{F}^2$. By the above reduction, from $\mathcal{A}$ we have a randomized algorithm $\mathcal{A}'$ that runs on the random matrix $X + \boldsymbol{G}$ with access to exact linear measurements and outputs a correct reconstruction $\hat{X}$ with probability $\geq 9/10$ if $X$ has rank $r$ and $\|X\|_\mathsf{F}^2 = \Theta(nr)$. Thus, for all such $X$,

$$\Pr_{\boldsymbol{G}, \boldsymbol{\sigma}}[\mathcal{A}'(X + \boldsymbol{G}, \boldsymbol{\sigma}) \text{ is correct}] \geq 9/10.$$

Here $\boldsymbol{\sigma}$ denotes the randomness used by the algorithm $\mathcal{A}'$. Now, if $\boldsymbol{X}$ is a random matrix constructed as $(\alpha/\sqrt{n})\sum_{i=1}^{r}\boldsymbol{u}_i\boldsymbol{v}_i^{\mathsf{T}}$ with $\boldsymbol{u}_i, \boldsymbol{v}_i$ being random vectors with independent Gaussian entries of mean 0 and variance 1, then with probability $\geq 99/100$, $\|\boldsymbol{X}\|_{\mathsf{F}}^2 = \Theta(nr)$. Thus, $\Pr_{\boldsymbol{X},\boldsymbol{G},\sigma}[\mathcal{A}'(\boldsymbol{X} + \boldsymbol{G}, \boldsymbol{\sigma})$ is correct$] \geq 8/10$. Hence, there exists some fixed $\sigma$ such that

$$\Pr_{\boldsymbol{X},\boldsymbol{G}}[\mathcal{A}'(\boldsymbol{X} + \boldsymbol{G}, \sigma) \text{ is correct}] \geq 8/10.$$

Thus, the existence of a randomized algorithm that solves low rank matrix recovery as in Theorem 1.1 implies the existence of a deterministic algorithm which given access to perfect linear measurements of random matrix $(\alpha/\sqrt{n})\sum_{i=1}^{r}\boldsymbol{u}_i\boldsymbol{v}_i^{\mathsf{T}} + \boldsymbol{G}$ outputs a reconstruction of $(\alpha/\sqrt{n})\sum_{i=1}^{r}\boldsymbol{u}_i\boldsymbol{v}_i^{\mathsf{T}}$ with probability $\geq 8/10$. This is essentially a reduction from randomized algorithms to deterministic algorithms using Yao's lemma. From here on, we prove lower bounds on such deterministic algorithms and conclude the lower bounds in Theorem 1.1. For simplicity, we explain the proof of our lower bound for $r = 1$ here and extend to general $r$ later. We consider the random matrix $\boldsymbol{G} + (\alpha/\sqrt{n})\boldsymbol{u}\boldsymbol{v}^{\mathsf{T}}$ and show how the lower bound proof proceeds.

Note that the matrix $\boldsymbol{A} := \boldsymbol{G} + (\alpha/\sqrt{n})\boldsymbol{u}\boldsymbol{v}^{\mathsf{T}} \in \mathbb{R}^{n \times n}$ can be flattened to the vector $\boldsymbol{a} = \boldsymbol{g} + (\alpha/\sqrt{n})(\boldsymbol{u} \otimes \boldsymbol{v}) \in \mathbb{R}^{n^2}$. Also, a general linear measurement, which we call a query $Q \in \mathbb{R}^{n \times n}$, can be vectorized to $q = \text{vec}(Q) \in \mathbb{R}^{n^2}$ with $Q(\boldsymbol{A}) = \langle q, \boldsymbol{a} \rangle$. Fix any deterministic algorithm. In the first round, the algorithm starts with a fixed matrix $Q^{(1)} \in \mathbb{R}^{k \times n^2}$ that corresponds to the $k$ queries and receives the response $\boldsymbol{r}^{(1)} := Q^{(1)}\boldsymbol{a}$. Then, as a function of the response $\boldsymbol{r}^{(1)}$ in the first round, the algorithm picks a matrix $Q^{(2)}[\boldsymbol{r}^{(1)}]$ in the second round and receives the response $\boldsymbol{r}^{(2)} := Q^{(2)}[\boldsymbol{r}^{(1)}] \cdot \boldsymbol{a}$. Similarly, in the $i$-th round, the deterministic algorithm picks a matrix $Q^{(i)}[\boldsymbol{r}^{(1)}, \ldots, \boldsymbol{r}^{(i-1)}] \in \mathbb{R}^{k \times n^2}$ as a function of $\boldsymbol{r}^{(1)}, \ldots, \boldsymbol{r}^{(i-1)}$ and receives the response $\boldsymbol{r}^{(i)} := Q^{(i)}[\boldsymbol{r}^{(1)}, \ldots, \boldsymbol{r}^{(i-1)}] \cdot \boldsymbol{a}$. Note that we assumed that the query matrices $Q^{(i)}$ chosen by the algorithm have orthonormal rows and also that $Q^{(i)}(Q^{(j)})^{\mathsf{T}} = 0$, i.e., the queries across rounds are also orthonormal.

For a fixed $u, v \in \mathbb{R}^n$, we see that the response $\boldsymbol{r}^{(1)} = Q^{(1)}\boldsymbol{g} + (\alpha/\sqrt{n})Q^{(1)}(u \otimes v)$. As the matrix $Q^{(1)}$ has orthonormal rows, the random variable $Q^{(1)}\boldsymbol{g} \equiv \boldsymbol{g}_k$, where $\boldsymbol{g}_k \sim N(0, I_k)$ is drawn from a mean-zero normal distribution with identity covariance. Thus, for fixed $u, v$, the distribution of the first round responses to the algorithm is $N((\alpha/\sqrt{n})Q^{(1)}(u \otimes v), I_k)$. Now the key observation is that $\|(\alpha/\sqrt{n})Q^{(1)}(\boldsymbol{u} \otimes \boldsymbol{v})\|_2^2 = \Theta(\alpha^2 k/n)$ with high probability over the inputs $(\boldsymbol{u}, \boldsymbol{v})$. This uses a recent concentration result for random tensors due to [31], and critically uses the fact that $Q^{(1)}$ has operator norm 1 and Frobenius norm $\sqrt{k}$. This means that for a large fraction of $(u, v)$ pairs, the distribution of the responses seen by the algorithm in the first round is close to $N(0, I_k)$, and therefore just looking at the response $\boldsymbol{r}^{(1)}$, the algorithm cannot have a lot of "information" about which $(u, v)$ pair is involved in the matrix that is unknown to the algorithm. So the query chosen in the second round $Q^{(2)}[\boldsymbol{r}^{(1)}]$ cannot have a "large" value of $Q^{(2)}[\boldsymbol{r}^{(1)}](u \otimes v)$ for most inputs $(u, v)$, with a high probability over the Gaussian component of the matrix.

Suppose the value of $Q^{(2)}[\boldsymbol{r}^{(1)}](u \otimes v)$ is small. Again, $\boldsymbol{r}^{(2)} = Q^{(2)}[\boldsymbol{r}^{(1)}]\boldsymbol{g} + (\alpha/\sqrt{n})Q^{(2)}[\boldsymbol{r}^{(1)}](u \otimes v)$. Crucially, as the queries in round 2 are orthogonal to the queries in round 1, we have by the rotational invariance of the Gaussian distribution that $Q^{(2)}[\boldsymbol{r}^{(1)}]\boldsymbol{g}$ is independent of $Q^{(1)}\boldsymbol{g}$, and that $Q^{(2)}[\boldsymbol{r}^{(1)}]\boldsymbol{g}$ is distributed as $N(0, I_k)$. So, for a fixed $u, v$, conditioned on the first round response $\boldsymbol{r}^{(1)}$, the distribution of the second round response $\boldsymbol{r}^{(2)}$ is given by $N((\alpha/\sqrt{n})Q^{(2)}[\boldsymbol{r}^{(1)}](u \otimes v), I_k) \approx N(0, I_k)$ using the assumption that $Q^{(2)}[\boldsymbol{r}^{(1)}](u \otimes v)$ is small. We again have that for a large fraction of pairs $(u, v)$ for which $Q^{(2)}[\boldsymbol{r}^{(1)}](u \otimes v)$ is small, the distribution of the second round response is also close to $N(0, I_k)$. The algorithm again does not gain a lot of information about which $(u, v)$ pair is involved in the matrix, and the third round query $Q^{(3)}[\boldsymbol{r}^{(1)}, \boldsymbol{r}^{(2)}]$ cannot have a large value of $\|Q^{(3)}[\boldsymbol{r}^{(1)}, \boldsymbol{r}^{(2)}](u \otimes v)\|_2$. The proof proceeds similarly for further rounds.

To formalize the above intuitive idea, we use Bayes risk lower bounds [7]. We show that with a large probability over the input matrix, the squared projection of $\boldsymbol{u} \otimes \boldsymbol{v}$ onto the query space of the algorithm is upper bounded and we use an iterative argument to show that an upper bound on the information up until round $i$ can in turn be used to upper bound the information up until round $i + 1$. Bayes risk bounds are very general and let us obtain upper bounds on the information learned by a deterministic learner. Concretely, let $\Theta$ be a parameter space and $\mathcal{P} = \{P_\theta : \theta \in \Theta\}$ be a set

of distributions, one for each $\theta \in \Theta$. Let $w$ be a distribution over $\Theta$. We sample $\boldsymbol{\theta} \sim w$ and then $\boldsymbol{x} \sim P_{\boldsymbol{\theta}}$ and provide the learner with $\boldsymbol{x}$. Given an action space $\mathcal{A}$, the learner uses a deterministic decision rule $\mathfrak{d} : \mathcal{X} \to \mathcal{A}$ to minimize a 0-1 loss function $L : \Theta \times \mathcal{A} \to \{0, 1\}$ in expectation, i.e., $\mathrm{E}_{\boldsymbol{\theta} \sim w}[\mathrm{E}_{\boldsymbol{x} \sim P_{\boldsymbol{\theta}}} L(\boldsymbol{\theta}, \mathfrak{d}(\boldsymbol{x}))]$. Let $R_{\mathrm{Bayes}}(L, \Theta, w) = \inf_{\mathfrak{d}} \mathrm{E}_{\boldsymbol{\theta} \sim w}[\mathrm{E}_{\boldsymbol{x} \sim P_{\boldsymbol{\theta}}} L(\boldsymbol{\theta}, \mathfrak{d}(\boldsymbol{x}))]$ be the loss achievable by the best deterministic decision rule $\mathfrak{d}$. Bayes risk lower bounds let us obtain lower bounds on $R_{\mathrm{Bayes}}$.

In our setting after round 1, we have $\Theta = \{ (u, v) : u, v \in \mathbb{R}^n \}$, $w$ is the joint distribution of two independent standard Gaussian random variables in $\mathbb{R}^n$ and for each $(u, v) \in \Theta$ we let $P_{uv}$ be the distribution of $\boldsymbol{r}^{(1)} = \boldsymbol{g}_k + (\alpha/\sqrt{n})Q^{(1)}(u \otimes v)$, an action space $\mathcal{A}$ of all $k \times n^2$ matrices with orthonormal rows (corresponding to the queries in the next round), and define a 0-1 loss function

$$L((u, v), Q) = \begin{cases} 0 & \text{if } \|Q(u \otimes v)\|_2^2 \geq T \\ 1 & \text{if } \|Q(u \otimes v)\|_2^2 < T \end{cases}$$

for an appropriate threshold parameter $T$. By setting $T$ appropriately as a function of $t$, we obtain a Bayes risk lower bound of $R_{\mathrm{Bayes}} \geq 1 - 1/(100t^2)$. Thus, we obtain that for any deterministic decision rule $\mathfrak{d}$, with probability $\geq 1 - 1/10t$ over $(\boldsymbol{u}, \boldsymbol{v}) \sim w$, we have

$$\mathrm{E}_{\boldsymbol{r}^{(1)} \sim P_{\boldsymbol{uv}}}[L((\boldsymbol{u}, \boldsymbol{v}), \mathfrak{d}(\boldsymbol{r}^{(1)}))] \geq 1 - 1/10t$$

and in particular, we have that with probability $\geq 1 - 1/10t$ over $(\boldsymbol{u}, \boldsymbol{v}) \sim w$,

$$\Pr_{\boldsymbol{r}^{(1)} \sim P_{\boldsymbol{uv}}} [\|Q^{(2)}[\boldsymbol{r}^{(1)}](\boldsymbol{u} \otimes \boldsymbol{v})\|_2^2 < T] \geq 1 - 1/10t. \tag{2}$$

The above statement essentially says that with probability $\geq 1 - 1/10t$ over the inputs $(\boldsymbol{u}, \boldsymbol{v})$, the second query $Q^{(2)}[\boldsymbol{r}^{(1)}]$ chosen by the deterministic algorithm has the property that $\|Q^{(2)}[\boldsymbol{r}^{(1)}](\boldsymbol{u} \otimes \boldsymbol{v})\|_2^2 < T$ with probability $\geq 1 - 1/10t$ over the Gaussian $\boldsymbol{G}$. In the second round, we restrict our analysis of the algorithm to only those $(u, v)$ which satisfy (2). We again define a distribution $w'$ over the inputs and for each such $(u, v)$ we define a distribution $P_{uv}$ over the round 1 and round 2 responses received by the algorithm. We define a new loss function with parameter $T' = \Delta \cdot T$ for a multiplicative factor $\Delta$ and again obtain a statement similar to (2) for a large fraction of inputs $(u, v)$ and repeat a similar argument for $t$ rounds and show that there is an $\Omega(1)$ fraction of the inputs for which the squared norm of the projection of $u \otimes v$ onto the query space after $t$ rounds is bounded by $T(t)$ with high probability over the Gaussian part of the input. This gives the result in Theorem 3.2. Note that $\|\boldsymbol{u} \otimes \boldsymbol{v}\|_2^2 = \Omega(n^2)$ with high probability and for any fixed matrix $Q$ with $k$ orthonormal rows, $\mathrm{E}[\|Q(\boldsymbol{u} \otimes \boldsymbol{v})\|_2^2] = k$ which corresponds to the amount of "information" the algorithm starts with. As we show that the "information" in each round grows by some multiplicative factor $\Delta$, a number $\Omega(\log_{\Delta}(n^2/k))$ of rounds is required to obtain an "information" of $\Theta(n^2)$, which is how we obtain our lower bounds. Here information is measured as the squared projection of $\boldsymbol{u} \otimes \boldsymbol{v}$ onto the query space of the algorithm.

We also extend our results to identifying a rank $r$ spike (sum of $r$ random outer products) corrupted by Gaussian noise. Specifically, we consider the random matrix $\boldsymbol{M} = \boldsymbol{G} + (\alpha/\sqrt{n})\sum_{i=1}^r \boldsymbol{u}_i\boldsymbol{v}_i^{\mathsf{T}}$ where all the coordinates of $\boldsymbol{G}, \boldsymbol{u}_i, \boldsymbol{v}_i$ are independently sampled from $N(0, 1)$. We show that if there is an algorithm that uses $k$ iterations in each round and identifies the spike with probability $\geq 9/10$, then it must run for $\Omega(\log(n^2/k)/\log \log n)$ rounds by appealing to the lower bound we described above for the case of $r = 1$. We show that if there is an algorithm for $r > 1$ that requires $t < O(\log(n^2/k)/\log \log n)$ adaptive rounds, then it can be used to solve the rank 1 spike estimation problem in $t$ rounds as well which contradicts the lower bound.

We then provide lower bounds on approximate algorithms for a host of problems such as spectral norm low rank approximation (LRA), Schatten norm LRA, Ky-Fan norm LRA and reduced rank regression, by showing that algorithms to solve these problems can be used to estimate the spike $\boldsymbol{uv}^{\mathsf{T}}$ in the random matrix $\boldsymbol{G} + (\alpha/\sqrt{n})\boldsymbol{uv}^{\mathsf{T}}$, and then use the aforementioned lower bounds on algorithms that can estimate the spike. Although our hard distribution is supported on non-symmetric matrices, we are still able to obtain lower bounds for algorithms for spectral norm LRA (rank $r \geq 2$) for symmetric matrices as well by considering a suitably defined symmetric random matrix using our hard distribution. Let $\boldsymbol{A} := \boldsymbol{G} + (\alpha/\sqrt{n})\boldsymbol{uv}^{\mathsf{T}}$ be the hard distribution in the case of rank $r = 1$.

We symmetrize the matrix by considering, $\boldsymbol{A}_{\mathrm{sym}} = \begin{bmatrix} 0 & \boldsymbol{A} \\ \boldsymbol{A}^{\mathsf{T}} & 0 \end{bmatrix}$. This symmetrization, as opposed

to $\boldsymbol{A}\boldsymbol{A}^{\mathsf{T}}$ or $\boldsymbol{A}^{\mathsf{T}}\boldsymbol{A}$, has the advantage that a linear measurement of $\boldsymbol{A}_{\mathrm{sym}}$ can be obtained from an appropriately transformed linear measurement of $\boldsymbol{A}$, thereby letting us obtain lower bounds for symmetric instances as well in the linear measurements model. However, we cannot obtain lower bounds for rank 1 spectral norm LRA for symmetric matrices using this symmetrization as the top two singular values of $\boldsymbol{A}_{\mathrm{sym}}$ are equal and hence even a zero matrix is a perfect rank 1 spectral norm LRA for $\boldsymbol{A}_{\mathrm{sym}}$ which does not give any information about the plant $\boldsymbol{u} \otimes \boldsymbol{v}$.

## 1.4   Related Work

As discussed, low rank matrix recovery has been extensively studied (see [5] and references therein for earlier work). Relatedly, [13] study the Robust PCA problem where the aim is to estimate the sum of a low rank matrix and a sparse matrix from noisy vector products with the hidden matrix. [30] study the Robust PCA problem when given access to linear measurements with the hidden matrix.

For non-adaptive algorithms for low rank matrix recovery with Gaussian errors, [5] show that their selectors based on the restricted isometry property of measurement matrices are optimal up to constant factors in the minimax error sense when the noise follows a Gaussian distribution. Our Theorem 1.1 extends their lower bounds and shows that if there is any randomized measurement matrix[2] $\boldsymbol{M}$ with $k$ rows coupled with a recovery algorithm that outputs a reconstruction for any rank $r$ matrix with $\|A\|_{\mathsf{F}}^2 = \Theta(nr)$, then it must have $k = \Omega(n^{2-o(1)})$. We again note that we give lower bounds even for algorithms with multiple adaptive rounds.

Our technique to show lower bounds is to plant a low rank matrix $(\alpha/\sqrt{n}) \sum_{i=1}^r u_i v_i^{\mathsf{T}}$ in an $n \times n$ Gaussian matrix so that any "orthonormal" access to the plant is corrupted by independent Gaussian noise. Notably this technique has been employed to obtain lower bounds on adaptive algorithms for sparse recovery in [23]. Even in the non-adaptive setting, Li and Woodruff [18] use the same hard distribution as we do to obtain sketching lower bounds for approximating Schatten $p$ norms, the operator norm, and the Ky Fan norms. The technique to show their lower bounds is that if a sketching matrix has $k \le c/(r^2 s^4)$ rows[3], it cannot distinguish between the random matrix $\boldsymbol{G}$ and the random matrix $\boldsymbol{G} + s \sum_{i=1}^r \boldsymbol{u}_i \boldsymbol{v}_i^{\mathsf{T}}$ where all the coordinates of $\boldsymbol{G}, \boldsymbol{u}_i, \boldsymbol{v}_i$ are drawn uniformly at random. They prove this by showing that $d_{\mathrm{TV}}(\mathcal{L}_1, \mathcal{L}_2)$ is small if $\mathcal{L}_1$ is the distribution of $M \cdot \mathrm{vec}(\boldsymbol{G})$ and $\mathcal{L}_2$ is the distribution of $M \mathrm{vec}(\boldsymbol{G} + s \sum_{i=1}^n \boldsymbol{u}_i \boldsymbol{v}_i^{\mathsf{T}})$ for any fixed measurement matrix $M$ with $k \le c/(r^2 s^4)$ rows. Their techniques do not extend to the plant estimation in the distribution $\boldsymbol{G} + s \sum_{i=1}^r \boldsymbol{u}_i \boldsymbol{v}_i^{\mathsf{T}}$ as the statement they prove only says that over the randomness of $\boldsymbol{G}, \boldsymbol{u}_i, \boldsymbol{v}_i$, the response distribution for $\boldsymbol{G} + s \sum_{i=1}^r \boldsymbol{u}_i \boldsymbol{v}_i^{\mathsf{T}}$ is close to the response distribution for $\boldsymbol{G}$, but in our case, we want the distributions $\mathcal{L}_{u,v}$ and $\mathcal{L}_{u',v'}$ to be indistinguishable, where $\mathcal{L}_{u,v}$ is the response distribution for $\boldsymbol{G} + (\alpha/\sqrt{n}) \sum_{i=1} u_i v_i^{\mathsf{T}}$.

Later, [28] considered the distribution of the symmetric random matrix $\boldsymbol{W} + \lambda \boldsymbol{U}\boldsymbol{U}^{\mathsf{T}}$, where $\boldsymbol{W}$ is the $n \times n$ symmetric Wigner matrix $(\boldsymbol{G} + \boldsymbol{G}^{\mathsf{T}})/\sqrt{2}$ and $\boldsymbol{U}$ is a uniformly random $n \times r$ matrix with orthonormal columns. They focus on obtaining lower bounds on adaptive algorithms that estimate the spike $\boldsymbol{U}$ in the matrix-vector product model. In particular, they show that if $Q$ is a basis for the query space spanned by any deterministic algorithm after querying $k$ adaptive matrix-vector queries, then $\lambda_r(Q^{\mathsf{T}}\boldsymbol{U}\boldsymbol{U}^{\mathsf{T}}Q)$ grows as $\sim \lambda^{k/r}$. Using this, they show that any algorithm which, given access to an arbitrary symmetric matrix $A$ in the matrix-vector product query model, must use $\Omega(r \log(n)/\sqrt{\mathrm{gap}})$ adaptive queries to output an $n \times r$ orthonormal matrix $V$ satisfying, for a small enough constant $c$,

$$\langle V, AV \rangle \ge (1 - c \cdot \mathrm{gap}) \sum_{i=1}^r \lambda_i(A), \tag{3}$$

where $\mathrm{gap} = (\lambda_r(A) - \lambda_{r+1}(A))/\lambda_r(A)$. We note the above guarantee is non-standard in the low rank approximation literature, which instead focuses more on approximation algorithms for Frobenius norm and spectral norm LRA. While in the matrix-vector product model, their hard distribution helps in getting lower bounds for numerical linear algebra problems on symmetric matrices, it seems that our hard distribution $\boldsymbol{G} + (\alpha/\sqrt{n}) \sum_{i=1}^r \boldsymbol{u}_i \boldsymbol{v}_i^{\mathsf{T}}$ is easier to understand in the linear measurement model and gives the important property that the noise between rounds is independent, which is

---

[2]Note that we assume a measurement matrix has orthonormal rows and that each measurement is corrupted by Gaussian noise of variance 1

[3]Their lower bound is a bit more general but we state this formulation for simplicity.

what lets us reduce the matrix-recovery problem with noisy measurements to spike estimation in $\boldsymbol{G} + (\alpha/\sqrt{n})\sum_{i=1}^{r}\boldsymbol{u}_i\boldsymbol{v}_i^{\mathsf{T}}$.

Recently, Bakshi and Narayanan [1] obtained a tight lower bound for rank-1 spectral norm low rank approximation problem in the matrix-vector product model. They show that $\Omega(\log n/\sqrt{\varepsilon})$ matrix vector products are required to obtain a $1 + \varepsilon$ spectral norm low rank approximation. We stress that while our results are not for $1 + \varepsilon$ approximations, they hold in the stronger linear measurements model.

## 2 Notation and Preliminaries

Given an integer $n \geq 1$, we use $[n]$ to denote the set $\{1, \ldots, n\}$. For a vector $v \in \mathbb{R}^n$, $\|v\|_2 := (\sum_{i \in [n]} v_i^2)^{1/2}$ denotes the Euclidean norm. For a matrix $A \in \mathbb{R}^{n \times d}$, $\|A\|_{\mathsf{F}}$ denotes the Frobenius norm $(\sum_{i,j} A_{ij}^2)^{1/2}$ and $\|A\|_2$ denotes the spectral norm $\sup_{x \neq 0} \|Ax\|_2/\|x\|_2$. For $p \geq 2$, we use $\|A\|_{\mathsf{S}_p}$ to denote the Schatten-$p$ norm $(\sum_i \sigma_i^p)^{1/p}$ and for $p \in [\min(n, d)]$, we use $\|A\|_{\mathsf{F}_p}$ to denote the Ky-Fan $p$ norm $\sum_{i=1}^{p} \sigma_i$, where $\sigma_1, \ldots, \sigma_{\min(n,d)}$ denote the singular values of the matrix $A$. Given a parameter $k$, the matrix $[A]_k$ denotes the best rank $k$ approximation for matrix $A$ in Frobenius norm. By the Eckart-Young-Mirsky theorem, the best rank $k$ approximation under any unitarily invariant norm, which includes the spectral norm, Frobenius norm, Schatten norms and Ky-Fan norms, is given by truncating the Singular Value Decomposition of the matrix $A$ to top $k$ singular values. We describe the Bayes Risk Lower Bounds that we use to prove Lemma 3.1 in the Appendix. Our presentation is based on [7]. Due to space constraints, we have placed all proofs in the appendix.

## 3 Number of Linear Measurements Versus Number of Adaptive Rounds

We now state the main theorem which shows a lower bound on the number of adaptive rounds required for any deterministic algorithm to estimate the *plant* when the input is a random matrix $\boldsymbol{G} + (\alpha/\sqrt{n})\sum_{i=1}^{r}\boldsymbol{u}_i\boldsymbol{v}_i^{\mathsf{T}}$. We use this theorem to prove Theorem 1.1 and lower bounds for other numerical linear algebra problems.

We prove the lower bound for the rank-$r$ plant estimation by first proving a lower bound on the rank-1 plant estimation and then reducing the rank-1 recovery problem to rank-$r$ recovery problem. For the rank-1 recovery problem, we prove the following lemma:

**Lemma 3.1.** *Given an integer $n$, and parameters $\alpha \geq 10$ and $\gamma \geq 1$, define the $n \times n$ random matrix $\boldsymbol{M} = \boldsymbol{G} + s\boldsymbol{u}\boldsymbol{v}^{\mathsf{T}}$ for $s := \alpha/\sqrt{n}$, where the entries of $\boldsymbol{G}$, $\boldsymbol{u}$, $\boldsymbol{v}$ are drawn independently from the distribution $N(0, 1)$. Let Alg be any $t$-round deterministic algorithm that makes $k \geq n$ adaptive linear measurements in each round. Let $Q^{(j)} \in \mathbb{R}^{k \times n^2}$ denote the matrix of general linear queries made by the algorithm in round $j$ and $Q$ be a matrix with orthonormal rows such that $\mathrm{rowspan}(Q) = \mathrm{rowspan}(Q^{(1)}) + \ldots + \mathrm{rowspan}(Q^{(t)})$. Then for all $t$ such that $O(k \log(n)) \leq (K\alpha^2\gamma^2)^t k \leq O(n^2)$ for a universal constant $K$,*

$$\Pr_{\boldsymbol{M} = \boldsymbol{G} + s\boldsymbol{u}\boldsymbol{v}^{\mathsf{T}}}[\|Q(\boldsymbol{u} \otimes \boldsymbol{v})\|_2^2 \leq (3K)k(K\alpha^2\gamma^2)^t] \geq (1 - 1/(10\gamma))^t - 1/\mathrm{poly}(n).$$

Setting $\gamma = O(\log(n))$, the theorem shows that if $t \leq c \log(n^2/k)/(\log\log(n) + \log(\alpha))$ for a small enough constant $c$, then for any $t$-round deterministic algorithm,

$$\Pr_{\boldsymbol{M} = \boldsymbol{G} + s\boldsymbol{u}\boldsymbol{v}^{\mathsf{T}}}[\|Q(\boldsymbol{u} \otimes \boldsymbol{v})\|_2^2 \leq n^2/100] \geq 4/5. \tag{4}$$

Setting $\gamma = O(1)$, we obtain that if $t \leq c \log(n^2/k)/(\log(\alpha))$ for a small enough constant $c$, then

$$\Pr_{\boldsymbol{M} = \boldsymbol{G} + s\boldsymbol{u}\boldsymbol{v}^{\mathsf{T}}}[\|Q(\boldsymbol{u} \otimes \boldsymbol{v})\|_2^2 \leq n^2/100] \geq 1/\mathrm{poly}(n). \tag{5}$$

The proof of the above lemma is in Appendix B. The above lemma directly shows lower bounds on any deterministic algorithm that can approximate the rank-1 planted matrix. We show that the lower bounds can be extended to algorithms that estimate the rank-$r$ plant as well.

**Theorem 3.2.** *Let $n$ and $r \leq n/2$ be input parameters and $\alpha$ be a large enough constant. Let the random matrix $\boldsymbol{G} + (\alpha/\sqrt{n}) \sum_{i=1}^{r} \boldsymbol{u}_i \boldsymbol{v}_i^{\mathsf{T}}$ be the input which can be accessed using linear measurements. If Alg is a $t$-round adaptive algorithm that uses $k$ linear measurements and at the end of $t$-rounds outputs a matrix $\hat{A}$ such that with probability $\geq 9/10$ over the randomness of the input and the internal randomness of the algorithm,*

$$\|\hat{A} - \sum_{i=1}^{r} \boldsymbol{u}_i \boldsymbol{v}_i^{\mathsf{T}}\|_{\mathsf{F}}^2 \leq c(n^2 r)$$

*for a small enough constant $c$, then $t = \Omega(\log(n^2/k)/(\log(\alpha) + \log\log n))$.*

Using the reduction from matrix recovery with noisy measurements to plant estimation with exact measurements that we described in the previous sections, we can now prove Theorem 1.1. The proof of Theorem 1.1 is in Appendix D.

## Acknowledgements

Praneeth Kacham and David P. Woodruff acknowledge the partial support from a Simons Investigator Award.

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

## A Preliminaries

### A.1 Bayes Risk Lower Bounds

Let $\Theta$ be a set of parameters and $\mathcal{P} = \{ P_\theta : \theta \in \Theta \}$ be a set of distributions over $\mathcal{X}$. Let $w$ be the prior distribution on $\Theta$ known to a learner. Suppose $\theta \sim w$, and $x \sim P_\theta$ and that the learner is given $x$. Now the learner wants to learn some information about $\theta$. Let $\mathfrak{d} : \mathcal{X} \to \mathcal{A}$ be a mapping from sample $x$ to action space $\mathcal{A}$ and $L : \Theta \times \mathcal{A} \to \{ 0, 1 \}$ be a zero-one loss function. We define

$$R_{\text{Bayes}}(L, \Theta, w) = \inf_{\mathfrak{d}} \int_\Theta \mathrm{E}_{x \sim P_\theta}[L(\theta, \mathfrak{d}(x))]w(d\theta) = \inf_{\mathfrak{d}} E_{\theta \sim w}[\mathrm{E}_{x \sim P_\theta} L(\theta, \mathfrak{d}(x))],$$

and

$$R_0(L, \Theta, w) = \inf_{a \in \mathcal{A}} \int_\Theta L(\theta, a)w(d\theta) = \inf_{a \in \mathcal{A}} \mathrm{E}_{\theta \sim w}[L(\theta, a)].$$

We drop $\Theta$ from the notation when it is clear.

$f$**-divergence** Let $\mathcal{C}$ denote the set of all convex functions $f$ with $f(1) = 0$. Let $P$ and $Q$ be two distributions with densities $p$ and $q$ over a common measure $\mu$. Then we have the $f$-divergence $D_f(P\|Q)$ defined as

$$D_f(P\|Q) = \int f\left(\frac{p}{q}\right) qd\mu + f'(\infty)P \{ q = 0 \}$$

using the convention $0 \cdot \infty = \infty$. For $f = x\log(x)$, the $f$-divergence $D_f(P\|Q) = d_{\text{KL}}(P\|Q)$, the Kullback–Leibler (KL) divergence. We frequently use $d_{\text{KL}}(\boldsymbol{X}\|\boldsymbol{Y})$ for some random variables $\boldsymbol{X}$ and $\boldsymbol{Y}$, which means the KL divergence between the distributions of $\boldsymbol{X}$ and $\boldsymbol{Y}$.

Now we can define the $f$-informativity of a set of distributions $\mathcal{P}$ with respect to a distribution $w$ as follows:

$$I_f(\mathcal{P}, w) = \inf_Q \int D_f(P_\theta\|Q)w(d\theta) = \inf_Q \mathrm{E}_{\theta \sim w}[D_f(P_\theta\|Q)].$$

We use $I(\mathcal{P}, w)$ to denote the $f$-informativity for $f = x\log x$. We finally have the following theorem from [7].

**Theorem A.1.** *For any 0-1 loss function L,*

$$I_f(\mathcal{P}, w) \geq \phi_f(R_{\text{Bayes}}, R_0).$$

*where for $0 \leq a, b \leq 1$, $\phi_f(a, b)$ denotes the $f$-divergence between distributions $P, Q$ over $\{ 0, 1 \}$ with $P \{ 1 \} = a$ and $Q \{ 1 \} = b$.*

As a corollary of the above theorem, [7] prove the following result which is the generalized Fano inequality.

**Theorem A.2.** *For any prior $w$ and 0-1 loss function L,*

$$R_{\text{Bayes}} \geq 1 + \frac{I(\mathcal{P}, w) + \log(1 + R_0)}{\log(1 - R_0)}.$$

### A.2 KL Divergence

We state some properties of the KL-divergence that we use throughout the paper.

**Lemma A.3.** *Let $P$ and $Q$ be two random variables with probability measures $p$ and $q$ such that $p$ is absolutely continuous with respect to $q$. Let $\tilde{P}$, with probability measure $\tilde{p}$, be the restriction of $P$ to an event $\mathcal{E}$, i.e., for all $\mathcal{E}'$, $\Pr[\tilde{P} \in \mathcal{E}'] = \Pr[P \in \mathcal{E} \cap \mathcal{E}']/\Pr(P \in \mathcal{E})$. Then*

$$d_{\text{KL}}(\tilde{p}\|q) \leq \frac{1}{\Pr(P \in \mathcal{E})}(d_{\text{KL}}(p\|q) + 2).$$

*Proof.* For $x \in \mathcal{E}$, we have $\tilde{p}(x) = p(x)/\Pr[P \in \mathcal{E}]$ and for $x \notin \mathcal{E}$, $\tilde{p}(x) = 0$. By definition

$$d_{\mathrm{KL}}(\tilde{p}\|q) = \int \tilde{p}(x) \log\left(\frac{\tilde{p}(x)}{q(x)}\right) \mathrm{d}x$$

using the convention $0 \cdot \log(0/1) = 0$. Now,

$$d_{\mathrm{KL}}(\tilde{p}\|q) = \int_{\mathcal{E}} \frac{p(x)}{\Pr(P \in \mathcal{E})} \log\left(\frac{p(x)}{\Pr(P \in \mathcal{E})q(x)}\right) \mathrm{d}x$$

$$= \frac{1}{\Pr(P \in \mathcal{E})} \int_{\mathcal{E}} p(x) \log\left(\frac{p(x)}{q(x)}\right) \mathrm{d}x + \log(1/\Pr(P \in \mathcal{E})).$$

We also have,

$$d_{\mathrm{KL}}(p\|q) = \int_{\mathcal{E}} p(x) \log\left(\frac{p(x)}{q(x)}\right) \mathrm{d}x + \int_{\bar{\mathcal{E}}} p(x) \log\left(\frac{p(x)}{q(x)}\right) \mathrm{d}x$$

which implies

$$\int_{\mathcal{E}} p(x) \log\left(\frac{p(x)}{q(x)}\right) \mathrm{d}x = d_{\mathrm{KL}}(p\|q) - \int_{\bar{\mathcal{E}}} p(x) \log\left(\frac{p(x)}{q(x)}\right) \mathrm{d}x.$$

Thus, it is enough to lower bound $\int_{\bar{\mathcal{E}}} p(x) \log\left(p(x)/q(x)\right) \mathrm{d}x$ to obtain an upper bound on the left hand side. So,

$$\int_{\bar{\mathcal{E}}} p(x) \log\left(\frac{p(x)}{q(x)}\right) \mathrm{d}x = -\Pr(P \in \bar{\mathcal{E}}) \int_{\bar{\mathcal{E}}} \frac{p(x)}{\Pr(P \in \bar{\mathcal{E}})} \log\left(\frac{q(x)}{p(x)}\right) \mathrm{d}x$$

$$\geq -\Pr(P \in \bar{\mathcal{E}}) \log\left(\frac{\Pr(Q \in \bar{\mathcal{E}})}{\Pr(P \in \bar{\mathcal{E}})}\right)$$

where the last inequality follows from $-\log(x)$ being convex. Finally,

$$d_{\mathrm{KL}}(\tilde{p}, q) = \frac{1}{\Pr(P \in \mathcal{E})} \int_{x \in \mathcal{E}} p(x) \log\left(\frac{p(x)}{q(x)}\right) \mathrm{d}x + \log(1/\Pr(P \in \mathcal{E}))$$

$$\leq \frac{1}{\Pr(P \in \mathcal{E})} d_{\mathrm{KL}}(p, q) + \frac{\Pr(P \in \bar{\mathcal{E}})}{\Pr(P \in \mathcal{E})} \log(\Pr(Q \in \bar{\mathcal{E}})/\Pr(P \in \bar{\mathcal{E}})) + \log(1/\Pr(P \in \mathcal{E}))$$

$$\leq \frac{1}{\Pr(P \in \mathcal{E})} d_{\mathrm{KL}}(p, q) + \frac{1}{\Pr(P \in \mathcal{E})} + \log(1/\Pr(P \in \mathcal{E}))$$

where we used $p \log(q/p) = p \log(q) + p \log(1/p) \leq 0 + 1 = 1$ for $0 \leq p, q \leq 1$. $\qquad\square$

The following lemma states the chain rule for KL-divergence.

**Lemma A.4** (Chain Rule). *Let $(X, Y)$ be jointly distributed random variables and $Z, W$ be independent random variables. Then*

$$d_{\mathrm{KL}}((X, Y)\|(Z, W)) = d_{\mathrm{KL}}(X\|Z) + \mathrm{E}_X[d_{\mathrm{KL}}((Y|X)\|W)].$$

The following lemma states the KL-divergence between two multivariate Gaussians.

**Lemma A.5** (KL-divergence between Gaussians, folklore).

$$d_{\mathrm{KL}}(N(\mu_1, \Sigma_1)\|N(\mu_2, \Sigma_2)) = \frac{1}{2}[\log \frac{\det(\Sigma_2)}{\det(\Sigma_1)} + \mathrm{tr}(\Sigma_2^{-1}\Sigma_1) + (\mu_2 - \mu_1)^{\mathsf{T}}\Sigma_2^{-1}(\mu_2 - \mu_1) - n]$$

*where $n$ is the dimension of the Gaussian.*

In this article, we use the above lemma only for $\Sigma_1 = \Sigma_2 = I$ in which case the above lemma implies that $d_{\mathrm{KL}}(N(\mu_1, I)\|N(\mu_2, I)) = (1/2)\|\mu_2 - \mu_1\|_2^2$.

## A.3 Properties of Gaussian random matrices and vectors

We use the following bounds on the norms of Gaussian matrices and vectors throughout the paper: (i) If $g$ is an $n$-dimensional Gaussian with all its entries independently drawn from $N(0,1)$, then with probability $\geq 1 - \exp(-\Theta(n))$, $(n/2) \leq \|g\|_2^2 \leq (3n/2)$, and (ii) If $G$ is an $m \times n$ $(m \geq n)$ Gaussian matrix with i.i.d. entries from $N(0,1)$, then $\Pr[\|G\|_2 \in [\sqrt{m} - \sqrt{n} - t, \sqrt{m} + \sqrt{n} + t]] \geq 1 - \exp(-\Theta(t^2))$. See [26, 17] and references therein for proofs.

**Rotational Invariance** We frequently use the rotational invariance property of multivariate Gaussian vectors $g \sim N(0, I_n)$, which implies that for any $n \times n$ orthogonal matrix $Q$ independent of $g$, the vector $Qg$ is also distributed as $N(0, I_n)$. Additionally, if $q_1, \dots, q_n$ denote rows of the matrix $Q$ with $q_1$ chosen arbitrarily independent of $g$ and the subsequent vectors $q_i := f_i(q_1, \dots, q_{i-1}, \langle q_1, g \rangle, \dots, \langle q_{i-1}, g \rangle)$ for arbitrary functions $f_i$ satisfying $q_i \perp q_1, \dots, q_{i-1}$, then $\langle q_i, g \rangle$ is distributed as $N(0,1)$ and is independent of $\langle q_1, g \rangle, \dots, \langle q_{i-1}, g \rangle$. We crucially use this property in the proof of Theorem 3.2.

# B   Proof of Lemma 3.1

First we discuss some notation, distributions and random variables that we use throughout the proof. Fix an arbitrary deterministic algorithm Alg. Let $A = G + suv^\mathsf{T}$ be a fixed realization of the random matrix $A$. Recall $s = \alpha/\sqrt{n}$. The algorithm Alg queries a fixed orthonormal matrix $Q^{(1)}$ and receives the response $r^{(1)} := Q^{(1)} \mathrm{vec}(A) = Q^{(1)}g + sQ^{(1)}(u \otimes v)$ where we use $g := \mathrm{vec}(G)$. As Alg is *deterministic*, in the second round it queries the matrix $Q^{(2)}[r^{(1)}]$. We use $Q^{(2)}[r^{(1)}]$ to emphasize that Alg picks $Q^{(2)}$ purely as a function of $r^{(1)}$. It receives the response $r^{(2)} = Q^{(2)}[r^{(1)}]g + sQ^{(2)}[r^{(1)}](u \otimes v)$ and picks queries $Q^{(3)}[r^{(1)}, r^{(2)}]$ for the third round and so on. Using $h^{(j)} := (r^{(1)}, \dots, r^{(j)})$ to denote the history of the algorithm until the end of round $j$, we have for $j = 0, \dots, t-1$

$$r^{(j+1)} := Q^{(j+1)}[h^{(j)}](g + s \cdot u \otimes v).$$

For each $j = 1, \dots, t$, the randomness of $A$ induces a distribution $H^{(j)}$ over histories until round $j$. Then for $u$ and $v$, the distribution $H_{uv}^{(j)} \equiv H^{(j)} | \{ u = u, v = v \}$ is the distribution of the history of Alg after $j$ rounds conditioned on $u = u$ and $v = v$ where the randomness in the histories is purely from the randomness of $G$. Recall that without loss of generality, we assume all the general linear queries made by the algorithm are orthogonal to each other. We first prove a tail bound on the amount of information that a non-adaptive query matrix can obtain.

## B.1   A Tail Bound

Let $u$ and $v$ be independent Gaussian random vectors. Let $Q \in \mathbb{R}^{k \times n^2}$ be an arbitrary matrix with $k$ orthonormal rows. We have the following lemma.

**Lemma B.1.** *There is a small enough universal constant $\beta$ such that for all $k \geq n$ and $C$ satisfying $4k \leq Ck \leq 16n^2$, we have for any orthonormal matrix $Q^{k \times n^2}$ that*

$$\Pr_{u,v}[\|Q \sum_i u_i \otimes v_i\|_2^2 \geq Ck] \leq \exp(-\beta Ck/n).$$

*Proof.* By Theorem 1.4 and Remark 1.5 of [31], we have that

$$\Pr_{u,v}[\|Q(u \otimes v)\|_2 \geq \|Q\|_\mathsf{F} + t] \leq \exp\left(-\frac{ct^2}{2n}\right)$$

for all $0 \leq t \leq 2n$. For $t = \sqrt{Ck/4}$, for $4k \leq Ck \leq 16n^2$, we have that

$$\Pr_{u,v}[\|Q(u \otimes v)\|_2 \geq \sqrt{Ck}] \leq \Pr_{u,v}[\|Q(u \otimes v)\|_2 \geq \sqrt{k} + \sqrt{Ck/4}]$$

$$\leq \exp\left(-\frac{cCk}{8n}\right)$$

which implies, taking $\beta = c/8$, that $\Pr_{\tilde{u},\tilde{v}}[\|Q(\tilde{u} \otimes v)\|_2^2 \geq Ck] \leq \exp(-\beta Ck/n)$. $\square$

## B.2   First Round

Let $w^{(1)}$ be the distribution of $(\boldsymbol{u}, \boldsymbol{v})$ where $\boldsymbol{u}$ and $\boldsymbol{v}$ are random variables that are independently sampled from $N(0, I_n)$. As already defined, the distribution of the history of the algorithm after the first round for a fixed $u, v$ is given by $H_{uv}^{(1)}$. We have that $H_{uv}^{(1)}$ is the distribution of the random variable

$$Q^{(1)}\boldsymbol{g} + sQ^{(1)}u \otimes v.$$

Let $P_{uv}^{(1)}$ be the distribution of the above random variable (although $P_{uv}^{(1)} \equiv H_{uv}^{(1)}$, we distinguish $P_{uv}^{(j)}$ and $H_{uv}^{(j)}$ in later rounds) and let $\mathcal{P}^{(1)} = \{\, P_{uv}^{(1)} \,\}$ be the set of distributions for all $u, v$. Then we have

$$I(\mathcal{P}^{(1)}, w^{(1)}) \leq \mathrm{E}_{(\boldsymbol{u},\boldsymbol{v})\sim w^{(1)}}[d_{\mathrm{KL}}(Q^{(1)}\boldsymbol{g} + sQ^{(1)}(\boldsymbol{u} \otimes \boldsymbol{v}) \,\|\, \boldsymbol{g}_k)]$$

$$\leq s^2 \, \mathrm{E}_{(\boldsymbol{u},\boldsymbol{v})\sim w^{(1)}} \|Q^{(1)}(\boldsymbol{u} \otimes \boldsymbol{v})\|_2^2 = (\alpha^2/n)kr.$$

We define the loss function

$$\mathrm{Loss}^{(1)}((u,v), Q) = \begin{cases} 0 & \text{if } \|Q(u \otimes v)\|_2^2 \geq f_1(\alpha, \gamma)k \\ 1 & \text{if } \|Q(u \otimes v)\|_2^2 < f_1(\alpha, \gamma)k \end{cases}$$

for $Q \in \mathbb{R}^{k \times n^2}$ with orthonormal rows and some function $f_1(\alpha, \gamma)$ satisfying $4 \leq f_1(\alpha, \gamma) \leq 16n^2/k$ for a parameter $\gamma \geq 1$. From Lemma B.1, we have

$$R_0(\mathrm{Loss}^{(1)}, w^{(1)}) \geq 1 - \exp(-\beta f_1(\alpha, \gamma)k/n)$$

which implies

$$R_{\mathrm{Bayes}} \geq 1 + \frac{(\alpha^2/n)k + \log(2)}{-\beta f_1(\alpha, \gamma)k/n + \log(2)}.$$

Picking $f_1(\alpha, \gamma) = K\alpha^2\gamma^2$, we have that $R_{\mathrm{Bayes}} \geq 1 - 1/(100\gamma^2)$. Thus with probability $1 - 1/(10\gamma)$ over $(\boldsymbol{u}, \boldsymbol{v}) \sim w^{(1)}$, we have that

$$\Pr_{\boldsymbol{h}^{(1)}\sim P_{\boldsymbol{uv}}^{(1)}}[\|Q^{(2)}[\boldsymbol{h}^{(1)}](\boldsymbol{u} \otimes \boldsymbol{v})\|_2^2 \leq f_1(\alpha, \gamma)k] \geq 1 - 1/(10\gamma).$$

Let $\mathrm{Good}^{(1)}$ be the set of all $(u, v)$ that satisfy the above property. Let $w^{(2)}$ be the distribution of $(\boldsymbol{u}, \boldsymbol{v}) \sim w^{(1)}$ conditioned on $(\boldsymbol{u}, \boldsymbol{v}) \in \mathrm{Good}^{(1)}$. For each $(u, v) \in \mathrm{Good}^{(1)}$, let

$$\mathrm{GoodH}_{uv}^{(1)} = \{\, h^{(1)} \,:\, \|Q^{(2)}[h^{(1)}](u \otimes v)\|_2^2 \leq f_1(\alpha, \gamma)k \,\}.$$

We have for all $(u, v) \in \mathrm{Good}^{(1)}$ that $\Pr_{\boldsymbol{h}^{(1)}\sim P_{uv}^{(1)}}[\boldsymbol{h}^{(1)} \in \mathrm{GoodH}_{uv}^{(1)}] \geq 1 - 1/(10\gamma)$. The overall conclusion of this is that with a large probability over $(\boldsymbol{u}, \boldsymbol{v}) \sim w^{(1)}$, the squared projection of $\boldsymbol{u} \otimes \boldsymbol{v}$ on the query asked by Alg in round 2 is small with large probability over $\boldsymbol{G}$.

## B.3   Further Rounds

We first define the following objects inductively.

1. For $j \geq 2$, let $w^{(j)}$ be the distribution of $(\boldsymbol{u}, \boldsymbol{v}) \sim w^{(j-1)}$ conditioned on $(\boldsymbol{u}, \boldsymbol{v}) \in \mathrm{Good}^{(j-1)}$. So $w^{(j)}$ is the distribution over only those inputs for which the squared projection of $u \otimes v$ on the query space of Alg until round $j$ is small with high probability over $\boldsymbol{G}$.

2. For $(u, v) \in \mathrm{Good}^{(j-1)}$,

   $$P_{uv}^{(j)} := \text{distribution of } \boldsymbol{h}^{(j)} = (\boldsymbol{h}^{(j-1)}, \boldsymbol{r}^{(j)}) \sim \boldsymbol{H}_{uv}^{(j)} \text{ conditioned on } \boldsymbol{h}^{(j-1)} \in \mathrm{GoodH}_{uv}^{(j-1)}.$$

   $P_{uv}^{(j)}$ denotes the distribution over histories after round $j$, conditioned on the queries used until round $j$ not having a lot of "information" about $u \otimes v$.

3. Let $\mathcal{P}^{(j)} := \{\, P_{uv}^{(j)} \,:\, (u, v) \in \mathrm{Good}^{(j-1)} \,\}$.

4. For $j \geq 1$,

$\text{Good}^{(j)} :=$

$\left\{ (u, v) \in \text{Good}^{(j-1)} \, : \, \Pr_{\boldsymbol{h}^{(j)} \sim P_{uv}^{(j)}}[\|Q^{(j+1)}[\boldsymbol{h}^{(j)}](u \otimes v)\|_2^2 \leq f_j(\alpha, \gamma)k] \geq 1 - 1/(10\gamma) \right\}.$

The set $\text{Good}^{(j)}$ denotes those values of $(u, v)$ for which with large probability over $\boldsymbol{G}$, the queries used by the algorithm until round $j + 1$ do not have a lot of "information" about $u \otimes v$.

5. For $(u, v) \in \text{Good}^{(j)}$,

$\text{GoodH}_{uv}^{(j)} :=$

$\left\{ h^{(j)} = (h^{(j-1)}, r^{(j)}) \, : \, h^{(j-1)} \in \text{GoodH}_{uv}^{(j-1)} \text{ and } \|Q^{(j+1)}[h^{(j)}](u \otimes v)\|_2^2 \leq f_j(\alpha, \gamma)k \right\}.$

$\text{GoodH}_{uv}^{(j)}$ denotes those histories, for which the queries used by the algorithm until round $j + 1$ do not have a lot of "information" about $u \otimes v$.

6. Let $f_0(\alpha, \gamma) = K$ and for $j \geq 1$, let $f_j(\alpha, \gamma) = K\alpha^2\gamma^2 f_{j-1}(\alpha, \gamma)$ for a large enough universal constant $K$.

**Lemma B.2.** *For all $1 \leq j$ satisfying $f_j(\alpha, \gamma)k \leq 16n^2$,*

*1.*

$$\Pr_{(\boldsymbol{u}, \boldsymbol{v}) \sim w^{(j)}}[(\boldsymbol{u}, \boldsymbol{v}) \in \text{Good}^{(j)}] \geq 1 - \frac{1}{10\gamma}$$

*2. For all $(u, v) \in \text{Good}^{(j)}$,*

$$\Pr_{\boldsymbol{h}^{(j)} \sim P_{uv}^{(j)}}[\boldsymbol{h}^{(j)} \in \text{GoodH}_{uv}^{(j)}] \geq 1 - \frac{1}{10\gamma}$$

*3.*

$$\mathrm{E}_{(\boldsymbol{u}, \boldsymbol{v}) \sim w^{(j)}}[d_{\text{KL}}(P_{\boldsymbol{u}\boldsymbol{v}}^{(j)} \,\|\, G^{(j)})] \leq f_j(\alpha, \gamma)(k/n),$$

*where $G^{(j)}$ is the joint distribution of $j$ independent $k$-dimensional Gaussian random variables.*

*Proof.* From the previous section, all the above statements hold for $j = 1$. Now assume that all the above statements hold for rounds $1, \ldots, j - 1$. We prove the statements inductively for round $j$. Recall $w^{(j)}$ is defined to be the distribution of $(\boldsymbol{u}, \boldsymbol{v}) \sim w^{(j-1)}$ conditioned on $(\boldsymbol{u}, \boldsymbol{v}) \in \text{Good}^{(j-1)}$. For each $(u, v) \in \text{Good}^{(j-1)}$, we now bound $d(P_{uv}^{(j)} \| G^{(j)})$. By definition,

$$d(P_{uv}^{(j)} \| G^{(j)}) = d_{\text{KL}}((\boldsymbol{h}^{(j-1)}, \boldsymbol{r}^{(j)}) \| (\boldsymbol{g}_k^{(1)}, \ldots, \boldsymbol{g}_k^{(j)}))$$

where $\boldsymbol{h}^{(j)} = (\boldsymbol{h}^{(j-1)}, \boldsymbol{r}^{(j)}) \sim P_{uv}^{(j)}$. Note that the marginal distribution of $\boldsymbol{h}^{(j-1)}$ is given by conditioning the distribution $P_{uv}^{(j-1)}$ on the event $\text{GoodH}_{uv}^{(j-1)}$. By Lemma A.4, we have

$d_{\text{KL}}((\boldsymbol{h}^{(j-1)}, \boldsymbol{r}^{(j)}) \| (\boldsymbol{g}_k^{(1)}, \ldots, \boldsymbol{g}_k^{(j)}))$
$= d_{\text{KL}}(\boldsymbol{h}^{(j-1)} \| (\boldsymbol{g}_k^{(1)}, \ldots, \boldsymbol{g}_k^{(j-1)})) + \mathrm{E}_{\boldsymbol{h}^{(j-1)}}[d_{\text{KL}}((\boldsymbol{r}^{(j)}) | \boldsymbol{h}^{(j-1)}) \| \boldsymbol{g}_k^{(j)}]$
$= d_{\text{KL}}((P_{uv}^{(j-1)} | \text{GoodH}_{uv}^{(j-1)}) \| G^{(j-1)}) + \mathrm{E}_{\boldsymbol{h}^{(j-1)} \sim P_{uv}^{(j-1)} | \text{GoodH}_{uv}^{(j-1)}}[d_{\text{KL}}((\boldsymbol{r}^{(j)}) | \boldsymbol{h}^{(j-1)}) \| \boldsymbol{g}_{nr}^{(j)}].$

$$\tag{6}$$

Now, using Lemma A.3, we have

$$d_{\text{KL}}((P_{uv}^{(j-1)} | \text{GoodH}_{uv}^{(j-1)}) \| G^{(j-1)}) \leq \frac{d_{\text{KL}}(P_{uv}^{(j-1)} \| G^{(j-1)}) + 2}{\Pr_{\boldsymbol{h}^{(j-1)} \sim P_{uv}^{(j-1)}}[\boldsymbol{h}^{(j-1)} \in \text{GoodH}^{(j-1)}]}$$

$$\leq (5/4)d_{\text{KL}}(P_{uv}^{(j-1)} \| G^{(j-1)}) + 5/2.$$

Here we used the inductive assumption that $\Pr_{\boldsymbol{h}^{(j-1)} \sim P_{uv}^{(j-1)}}[\boldsymbol{h}^{(j-1)} \in \mathrm{GoodH}^{(j-1)}] \geq 1 - 1/(10\gamma) \geq 9/10$ where the last inequality follows as the parameter $\gamma \geq 1$. Next, we upper bound the second term in (6). We have that $\boldsymbol{r}^{(j)}|\boldsymbol{h}^{(j-1)} = Q^{(j)}[\boldsymbol{h}^{(j-1)}]\boldsymbol{g} + s(Q^{(j)}[\boldsymbol{h}^{(j-1)}])(u \otimes v)$ is distributed as $N(s(Q^{(j)}[\boldsymbol{h}^{(j-1)}])(u \otimes v), I_k)$ by rotational invariance of the Gaussian distribution. Therefore,

$$d_{\mathrm{KL}}((\boldsymbol{r}^{(j)}|\boldsymbol{h}^{(j-1)})\|\boldsymbol{g}_k^{(j)}) = (1/2)s^2\|Q^{(j)}[\boldsymbol{h}^{(j-1)}](u \otimes v)\|_2^2 \leq (\alpha^2/n)f_{j-1}(\alpha,\gamma)k.$$

In the last inequality, we used the fact that $\boldsymbol{h}^{(j-1)} \in \mathrm{GoodH}_{uv}^{(j-1)}$. Finally,

$$\mathrm{E}_{(\boldsymbol{u},\boldsymbol{v})\sim w^{(j)}}[d(P_{\boldsymbol{uv}}^{(j)}\|G^{(j)})] \leq (5/4)\,\mathrm{E}_{(\boldsymbol{u},\boldsymbol{v})\sim w^{(j)}}\,d_{\mathrm{KL}}(P_{\boldsymbol{uv}}^{(j-1)}\|G^{(j-1)}) + 5/2 + \alpha^2 f_{j-1}(\alpha,\gamma)(k/n)$$

$$\leq (5/2)\,\mathrm{E}_{(\boldsymbol{u},\boldsymbol{v})\sim w^{(j-1)}}\,d_{\mathrm{KL}}(P_{\boldsymbol{uv}}^{(j-1)}\|G^{(j-1)}) + 5/2 + \alpha^2 f_{j-1}(\alpha,\gamma)(k/n)$$

as the distribution $w^{(j)}$ is obtained by conditioning $w^{(j-1)}$ on an event with probability $\geq 1 - 1/(10\gamma) \geq 1/2$ and $d_{\mathrm{KL}}(\cdot\|\cdot) \geq 0$. Now, using the inductive assumption, we have

$$\mathrm{E}_{(\boldsymbol{u},\boldsymbol{v})\sim w^{(j)}}[d(P_{\boldsymbol{uv}}^{(j)}\|G^{(j)})] \leq (5/2)f_{j-1}(\alpha,\gamma)(k/n) + 5/2 + \alpha^2 f_{j-1}(\alpha,\gamma)(k/n)$$

$$\leq 2\alpha^2 f_{j-1}(\alpha,\gamma)(k/n) \leq f_j(\alpha,\gamma)(k/n)$$

where we use $\alpha \geq 15$. This proves the third statement in the lemma for round $j$. We also have, $I(w^{(j)}, \mathcal{P}^{(j)}) \leq \mathrm{E}_{(\boldsymbol{u},\boldsymbol{v})\sim w^{(j)}}[d(P_{\boldsymbol{uv}}^{(j)}\|G^{(j)})] \leq 2\alpha^2 f_{j-1}(\alpha,\gamma)(k/n)$. We now define a loss function $L^{(j)}$ and use Bayes risk lower bounds to prove the remaining statements. Let

$$\mathrm{Loss}^{(j)}((u,v),Q) = \begin{cases} 1 & \text{if } \|Q(u \otimes v)\|_2^2 \leq f_j(\alpha,\gamma)k \\ 0 & \text{if } \|Q(u \otimes v)\|_2^2 > f_j(\alpha,\gamma)k \end{cases}$$

where $Q \in \mathbb{R}^{k \times n^2}$ is an orthonormal matrix with $k \geq n$ rows. We have for $j$ such that $f_j(\alpha,\gamma)k \leq 16n^2$,

$$R_0(\mathrm{Loss}^{(j)}, w^{(j)}) = \inf_Q \mathrm{E}_{(\boldsymbol{u},\boldsymbol{v})\sim w^{(j)}}[\mathrm{Loss}^{(j)}((\boldsymbol{u},\boldsymbol{v}),Q)] \geq 1 - (1-1/(10\gamma))^{-j}\exp(-\beta f_j(\alpha,\gamma)k/n)$$

where we use Lemma B.1 and the fact that the distribution $w^{(j)}$ is obtained by conditioning $w^{(1)}$ on an event with probability $\geq (1 - 1/(10\gamma))^j$ which we prove in section B.4. By the generalized Fano inequality, we obtain

$$R_{\mathrm{Bayes}}(\mathrm{Loss}^{(j)}, w^{(j)}) \geq 1 + \frac{I(w^{(j)}, \mathcal{P}^{(j)}) + \log(2)}{\log(1 - R_0(\mathrm{Loss}^{(j)}, w^{(j)}))}$$

$$\geq 1 - \frac{2\alpha^2 f_{j-1}(\alpha,\gamma)k/n + \log(2)}{\beta f_j(\alpha,\gamma)k/n + j\log(1 - 1/(10\gamma))}.$$

As $f_j(\alpha,\gamma) = K\alpha^2\gamma^2 f_{j-1}(\alpha,\gamma)$ and $f_0(\alpha,\gamma) \geq K$ for a large enough constant $K$, we have

$$\beta f_j(\alpha,\gamma) \geq -10j\log(1 - 1/(10\gamma))$$

and therefore for $K$ large enough,

$$R_{\mathrm{Bayes}}(\mathrm{Loss}^{(j)}, w^{(j)}) \geq 1 - \frac{1}{100\gamma^2}.$$

By definition of $R_{\mathrm{Bayes}}$, we conclude that

$$\mathrm{E}_{(\boldsymbol{u},\boldsymbol{v})\sim w^{(j)}}[\mathrm{E}_{\boldsymbol{h}^{(j)}\sim P_{\boldsymbol{uv}}^{(j)}}[\mathrm{Loss}((\boldsymbol{u},\boldsymbol{v}),Q^{(j+1)}[\boldsymbol{h}^{(j)}])]] \geq 1 - 1/(100\gamma^2).$$

By Markov's inequality, we have that with probability $\geq 1 - 1/(10\gamma)$ over $(\boldsymbol{u},\boldsymbol{v}) \sim w^{(j)}$, it holds that

$$\mathrm{E}_{\boldsymbol{h}^{(j)}\sim P_{\boldsymbol{uv}}^{(j)}}[\mathrm{Loss}((\boldsymbol{u},\boldsymbol{v}),Q^{(j+1)}[\boldsymbol{h}^{(j)}])] \geq 1 - 1/(10\gamma)$$

which is equivalent to

$$\Pr_{\boldsymbol{h}^{(j)}\sim P_{\boldsymbol{uv}}^{(j)}}[\|Q^{(j+1)}[\boldsymbol{h}^{(j)}](\boldsymbol{u} \otimes \boldsymbol{v})\|_2^2 \leq f_j(\alpha,\gamma)k] \geq 1 - 1/(10\gamma).$$

Thus, we conclude that $\Pr_{(\boldsymbol{u},\boldsymbol{v})\sim w^{(j)}}[(\boldsymbol{u},\boldsymbol{v}) \in \mathrm{Good}^{(j)}] \geq 1 - 1/(10\gamma)$ and that for $(u,v) \in \mathrm{Good}^{(j)}$, we have $\Pr_{\boldsymbol{h}^{(j)}\sim P_{\boldsymbol{uv}}^{(j)}}[\boldsymbol{h}^{(j)} \in \mathrm{GoodH}_{uv}^{(j)}] \geq 1 - 1/(10\gamma)$. $\qquad\square$

## B.4 Wrap-up

Let $j \geq 1$ satisfy $f_j(\alpha, \gamma)k \leq 16n^2$. By definition, $\mathrm{Good}^{(j)} \subseteq \mathrm{Good}^{(j-1)} \subseteq \ldots \mathrm{Good}^{(1)}$. We also have

$$w^{(j)} = w^{(j-1)}|\mathrm{Good}^{(j-1)} = w^{(1)}|\mathrm{Good}^{(j-1)}.$$

Now, using the fact that $\mathrm{Good}^{(j)} \subseteq \mathrm{Good}^{(j-1)}$,

$$\frac{\Pr_{(\boldsymbol{u},\boldsymbol{v})\sim w^{(1)}}[(\boldsymbol{u}, \boldsymbol{v}) \in \mathrm{Good}^{(j)}]}{\Pr_{(\boldsymbol{u},\boldsymbol{v})\sim w^{(1)}}[(\boldsymbol{u}, \boldsymbol{v}) \in \mathrm{Good}^{(j-1)}]} = \Pr_{(\boldsymbol{u},\boldsymbol{v})\sim w^{(1)}}[(\boldsymbol{u}, \boldsymbol{v}) \in \mathrm{Good}^{(j)}|(\boldsymbol{u}, \boldsymbol{v}) \in \mathrm{Good}^{(j-1)}]$$

As $w^{(j)} = w^{(1)}|\mathrm{Good}^{(j-1)}$, we have

$$\Pr_{(\boldsymbol{u},\boldsymbol{v})\sim w^{(1)}}[(\boldsymbol{u}, \boldsymbol{v}) \in \mathrm{Good}^{(j)}|(\boldsymbol{u}, \boldsymbol{v}) \in \mathrm{Good}^{(j-1)}] = \Pr_{(\boldsymbol{u},\boldsymbol{v})\sim w^{(j)}}[(\boldsymbol{u}, \boldsymbol{v}) \in \mathrm{Good}^{(j)}]$$
$$\geq (1 - 1/(10\gamma))$$

where the last inequality is from Lemma B.2. Thus, $\Pr_{(\boldsymbol{u},\boldsymbol{v})\sim w^{(1)}}[(\boldsymbol{u}, \boldsymbol{v}) \in \mathrm{Good}^{(j)}] \geq (1 - 1/(10\gamma))^j$.

Similarly, for $(u, v) \in \mathrm{Good}^{(j)}$, we have that

$$\frac{\Pr_{\boldsymbol{h}^{(j)}\sim H_{uv}^{(j)}}[\boldsymbol{h}^{(j)} \in \mathrm{GoodH}_{uv}^{(j)}]}{\Pr_{\boldsymbol{h}^{(j-1)}\sim H_{uv}^{(j-1)}}[\boldsymbol{h}^{(j-1)} \in \mathrm{GoodH}_{uv}^{(j-1)}]} = \Pr_{\boldsymbol{h}^{(j)}\sim P_{uv}^{(j)}}[\boldsymbol{h}^{(j)} \in \mathrm{GoodH}_{uv}^{(j)}]$$
$$\geq (1 - 1/(10\gamma))$$

where again, the last inequality follows from Lemma B.2. Thus, $\Pr_{\boldsymbol{h}^{(j)}\sim H_{uv}^{(j)}}[\boldsymbol{h}^{(j)} \in \mathrm{GoodH}_{uv}^{(j)}] \geq (1 - 1/(10\gamma))^j$. Now, for $(\boldsymbol{u}, \boldsymbol{v}) \sim w^{(1)}$ and $\boldsymbol{h}^{(j)} \sim H_{\boldsymbol{u}\boldsymbol{v}}^{(j)}$, we have that with probability $\geq (1 - 1/(10\gamma))^{2j}$ it holds that $(\boldsymbol{u}, \boldsymbol{v}) \in \mathrm{Good}^{(j)}$ and $\boldsymbol{h}^{(j)} \in \mathrm{GoodH}_{\boldsymbol{u}\boldsymbol{v}}^{(j)}$. Thus, with probability $\geq (1 - 1/(10\gamma))^{2j}$ over the input matrix $\boldsymbol{G} + \alpha \sum_{i=1}^r \boldsymbol{u}_i \boldsymbol{v}_i^\mathsf{T}$, we have that

$$\sum_{j'=1}^{j} \|Q^{(j'+1)}[\boldsymbol{h}^{(j')}](\boldsymbol{u} \otimes \boldsymbol{v})\|_2^2 \leq \sum_{j'=1}^{j} f_{j'}(\alpha, \gamma)k \leq 2f_j(\alpha, \gamma)k.$$

With probability $\geq 1 - 1/\operatorname{poly}(n)$, using Lemma B.1, $\|Q^{(0)}(\boldsymbol{u} \otimes \boldsymbol{v})\|_2^2 \leq k \log(n)$. Using a union bound, we obtain that with probability $\geq (1 - 1/(10\gamma))^{2j} - 1/\operatorname{poly}(n)$,

$$\sum_{j'=0}^{j} \|Q^{(j'+1)}[\boldsymbol{h}^{(j')}](\boldsymbol{u} \otimes \boldsymbol{v})\|_2^2 \leq 2f_j(\alpha, \gamma)kr + Kk \log(n)$$
$$= 2kf_0(\alpha, \gamma)(K\alpha^2\gamma^2)^j + Kk \log(n)$$
$$= (3K)k(K\alpha^2\gamma^2)^j$$

where $K$ is a large enough absolute constant which proves the theorem for $j$ such that $(K\alpha^2\gamma^2)^j = \Omega(\log(n))$.

## C  Proof of Theorem 3.2

Suppose Alg is a $t$ round algorithm that uses $k$ liner measurements in each round such that when run on the matrix $\boldsymbol{G} + (\alpha/\sqrt{n})\boldsymbol{u}\boldsymbol{v}^\mathsf{T}$, it produces a matrix $\hat{A}$ such that with probability $9/10$ over the input matrix,

$$\|\hat{A} - \boldsymbol{u}\boldsymbol{v}^\mathsf{T}\|_\mathsf{F}^2 \leq cn^2$$

for a small enough constant $c$. By making the algorithm to query the vector $\operatorname{vec}(\hat{A})$ in round $t + 1$, we can therefore ensure that if $Q$ is the overall query space of the algorithm, then

$$\Pr[\|Q(\boldsymbol{u} \otimes \boldsymbol{v})\|_2^2 \geq n^2/100] \geq 4/5.$$

By Lemma 3.1, we obtain that $t + 1 = \Omega(\log(n^2/k)/(\log(\alpha) + \log\log n))$ and hence $t = \Omega(\log(n^2/k)/(\log(\alpha) + \log\log n))$.

Now suppose $\mathsf{Alg}_r$ is a $t$ round algorithm that uses $k$ linear measurements of the input random matrix $\boldsymbol{G} + (\alpha/\sqrt{n}) \sum_{i=1}^r \boldsymbol{u}_i \boldsymbol{v}_i^\mathsf{T}$ and outputs a matrix $\hat{A}$ such that with probability $\geq 9/10$ over the input,

$$\|\hat{A} - \sum_{i=1}^r \boldsymbol{u}_i \boldsymbol{v}_i^\mathsf{T}\|_\mathsf{F}^2 \leq cn^2 r$$

for a small enough constant $c$. We obtain a lower bound on $t$ by reducing the rank-$1$ plant estimation problem to the rank-$r$ plant estimation problem. Suppose the input is the random matrix $\boldsymbol{G} + (\alpha/\sqrt{n})\boldsymbol{u}_1 \boldsymbol{v}_1^\mathsf{T}$. We sample $\boldsymbol{u}_2, \ldots, \boldsymbol{u}_r$ and $\boldsymbol{v}_2, \ldots, \boldsymbol{v}_r$ so that the coordinates of these vectors are independent standard Gaussian random variables. Now we note that we can perform arbitrary linear measurements of the matrix $\boldsymbol{G} + (\alpha/\sqrt{n}) \sum_{i=1}^r \boldsymbol{u}_i \boldsymbol{v}_i^\mathsf{T}$ since we have access to linear measurements of $\boldsymbol{G} + (\alpha/\sqrt{n})\boldsymbol{u}_1 \boldsymbol{v}_1^\mathsf{T}$ and we know the vectors $\boldsymbol{u}_2, \ldots, \boldsymbol{u}_r$ and $\boldsymbol{v}_2, \ldots, \boldsymbol{v}_r$. We can then use Algorithm $\mathsf{Alg}_r$ to obtain a matrix $\hat{A}$ such that with probability $\geq 9/10$ over the input and our sampled vectors,

$$\|\sum_{i=1}^r \boldsymbol{u}_i \boldsymbol{v}_i^\mathsf{T} - \hat{A}\|_\mathsf{F}^2 \leq cn^2 r.$$

Condition on this event. We have $\|\hat{A} - [\hat{A}]_r\|_\mathsf{F}^2 \leq \|\sum_{i=1}^r \boldsymbol{u}_i \boldsymbol{v}_i^\mathsf{T} - \hat{A}\|_\mathsf{F}^2 \leq cn^2 r$ which then implies $\|\sum_{i=1}^r \boldsymbol{u}_i \boldsymbol{v}_i^\mathsf{T} - [\hat{A}]_r\|_\mathsf{F}^2 \leq 2cn^2 r$. Now, let $P$ be the at most rank $r$ projection matrix onto the rowspace of the matrix $[\hat{A}]_r$ and assume that $\boldsymbol{U}$ and $\boldsymbol{V}$ are matrices with columns given by $\boldsymbol{u}_1, \ldots, \boldsymbol{u}_r$ and $\boldsymbol{v}_1, \ldots, \boldsymbol{v}_r$ respectively. From above, we get

$$\|\boldsymbol{U}\boldsymbol{V}^\mathsf{T}(I - P)\|_\mathsf{F}^2 \leq \|\boldsymbol{U}\boldsymbol{V}^\mathsf{T} - [\hat{A}]_r\|_\mathsf{F}^2 \leq 2cn^2 r$$

which then implies that

$$\sigma_{\min}(\boldsymbol{U})^2 \|\boldsymbol{V}^\mathsf{T}(I - P)\|_\mathsf{F}^2 \leq \|\boldsymbol{U}\boldsymbol{V}^\mathsf{T}(I - P)\|_\mathsf{F}^2 \leq 2cn^2 r.$$

Now, $\boldsymbol{U}$ is an $n \times r$ matrix with coordinates being independent standard Gaussian random variables. We have from [26] that if $r \leq n/2$, then with probability $1 - \exp(-n)$, $\sigma_{\min}(\boldsymbol{U}) \geq c_1 \sqrt{n}$. From [17] we also have that with probability $\geq 1 - \exp(-n)$, $c_2 n \leq \|\boldsymbol{u}_1\|_2^2, \ldots, \|\boldsymbol{u}_2\|_2^2 \leq Cn$ for a small enough $c_2$ and large enough $C$. Conditioned on all these events, we have

$$\sum_{i=1}^r \|\boldsymbol{u}_i \boldsymbol{v}_i^\mathsf{T}(I - P)\|_\mathsf{F}^2 \leq Cn \sum_{i=1}^r \|\boldsymbol{v}_i^\mathsf{T}(I - P)\|_2^2$$

$$\leq Cn \|\boldsymbol{V}^\mathsf{T}(I - P)\|_\mathsf{F}^2 \leq \frac{(2cn^2 r)(Cn)}{c_1^2 n} \leq (2cC/c_1)n^2.$$

Hence, at least $9r/10$ indices $i$ have the property that $\|\boldsymbol{u}_i \boldsymbol{v}_i^\mathsf{T}(I - P)\|_\mathsf{F}^2 \leq (20cC/c_1)n^2$. Since the marginal distributions of $\boldsymbol{u}_1 \boldsymbol{v}_1^\mathsf{T}, \ldots, \boldsymbol{u}_r \boldsymbol{v}_r^\mathsf{T}$ are identical, we obtain that with probability $\geq 8/10$,

$$\|\boldsymbol{u}_1 \boldsymbol{v}_1^\mathsf{T}(I - P)\|_\mathsf{F}^2 \leq (20cC/c_1)n^2.$$

Note that rank of $P$ is at most $r$ and therefore $P = QQ^\mathsf{T}$ for an orthonormal matrix $Q$ with at most $r$ columns. In the $(t + 1)$-th round, we make $nr$ linear measurements of the input matrix and obtain the matrix

$$\boldsymbol{G}Q + (\alpha/\sqrt{n})\boldsymbol{u}_1 \boldsymbol{v}_1^\mathsf{T}Q$$

and therefore, we can obtain the matrix $M = \boldsymbol{G}P + (\alpha/\sqrt{n})\boldsymbol{u}_1 \boldsymbol{v}_1^\mathsf{T}P$. Let $[M]_1$ be the best rank-$1$ approximation of $M$ in operator norm. We have

$$\|[M]_1 - (\alpha/\sqrt{n})\boldsymbol{u}_1 \boldsymbol{v}_1^\mathsf{T}\|_2 \leq \|[M]_1 - M\|_2 + \|M - (\alpha/\sqrt{n})\boldsymbol{u}_1 \boldsymbol{v}_1^\mathsf{T}P\|_2 + \|(\alpha/\sqrt{n})\boldsymbol{u}_1 \boldsymbol{v}_1^\mathsf{T}P - (\alpha/\sqrt{n})\boldsymbol{u}_1 \boldsymbol{v}_1^\mathsf{T}\|_2$$

$$\leq \|\boldsymbol{G}P\|_2 + \|\boldsymbol{G}P\|_2 + (\alpha/\sqrt{n})\sqrt{(20cC/c_1)n^2}.$$

Since, $\|\boldsymbol{G}\|_2 \leq 2\sqrt{n}$ with probability $1 - \exp(-\Theta(n))$, we get

$$\|(\sqrt{n}/\alpha)[M]_1 - \boldsymbol{u}_1 \boldsymbol{v}_1^\mathsf{T}\|_2 \leq (4/\alpha + \sqrt{20cC/c_1})n$$

and

$$\|(\sqrt{n}/\alpha)[M]_1 - \boldsymbol{u}_1\boldsymbol{v}_1^\mathsf{T}\|_\mathsf{F}^2 \le 2(4/\alpha + \sqrt{20cC/c_1})^2 n^2.$$

For $\alpha$ large enough and $c$ small enough, by querying the vector $\mathrm{vec}([M]_1)$, we can again ensure that if $Q$ is the query space of the algorithm after $t+2$ rounds, we have with probability $\ge 1/2$ over the input random matrix that

$$\|Q(\boldsymbol{u}_1 \otimes \boldsymbol{v}_1)\|_2^2 \ge n^2/100.$$

We again have from Lemma 3.1 that $t + 2 = \Omega(\log(n^2/k)/(\log\log n + \log\alpha))$ and therefore that $t = \Omega(\log(n^2/k)/(\log\log n + \log\alpha))$.

# D  Proof of Theorem 1.1 using Theorem 3.2

Let $\mathcal{A}$ be any any randomized algorithm that queries orthonormal measurements of the underlying matrix $A$ and outputs a reconstruction $\hat{A}$ of $A$ satisfying $\|A - \hat{A}\|_\mathsf{F}^2 \le c\|A\|_\mathsf{F}^2$ with probability $\ge p$, over both the randomness of the algorithm and randomness of the measurements. Let $\hat{A} = \mathcal{A}(A, \boldsymbol{\sigma}, \boldsymbol{\gamma})$ where $\boldsymbol{\sigma}$ captures the randomness of the algorithm and $\boldsymbol{\gamma}$ captures the randomness in measurements. First we have the following lemma.

**Lemma D.1.** *If there is a randomized algorithm $\mathcal{A}(A, \boldsymbol{\sigma}, \boldsymbol{\gamma})$ such that for all rank $\le r$ matrices $A$ with $\|A\|_\mathsf{F}^2 = \Theta(nr)$,*

$$\Pr_{\boldsymbol{\sigma},\boldsymbol{\gamma}}[\|\mathcal{A}(A, \boldsymbol{\sigma}, \boldsymbol{\gamma}) - A\|_\mathsf{F}^2 \le c\|A\|_\mathsf{F}^2] \ge p,$$

*then there is a randomized algorithm $\mathcal{A}'$ such that for all $A$ with $\|A\|_\mathsf{F}^2 = \Theta(nr)$,*

$$\Pr_{\boldsymbol{G},\boldsymbol{\sigma}}[\|\mathcal{A}'(A + \boldsymbol{G}, \boldsymbol{\sigma}) - A\|_\mathsf{F}^2 \le c\|A\|_\mathsf{F}^2] \ge p.$$

*Proof.* Fix a particular $\sigma$. The algorithm first queries an orthonormal matrix $Q^{(1)} \in \mathbb{R}^{k \times n^2}$ and receives a random response $\boldsymbol{r}^{(1)} = Q^{(1)} \cdot \mathrm{vec}(A) + \boldsymbol{g}^{(1)}$ where $\boldsymbol{g}^{(1)}$ is a vector with independent Gaussian components of mean 0 and variance 1. As a function of $\boldsymbol{r}^{(1)}$, the algorithm queries an orthonormal matrix $Q^{(1)}[\boldsymbol{r}^{(1)}]$ and receives the random response $\boldsymbol{r}^{(2)} = Q^{(2)}[\boldsymbol{r}^{(1)}] \cdot \mathrm{vec}(A) + \boldsymbol{g}^{(2)}$ where $\boldsymbol{g}^{(2)}$ is again a random Gaussian vector *independent* of $\boldsymbol{g}^{(1)}$. Importantly, we also have that $Q^{(2)}[\boldsymbol{r}^{(1)}]$ is orthogonal to the query matrix $Q^{(1)}$ in the first round. The algorithm proceeds in further rounds accordingly where it picks query matrices as a function of the responses in all the previous rounds.

Let $\boldsymbol{G}$ be an $n \times n$ Gaussian matrix with independent entries of mean 0 and variance 1. We now observe that $\boldsymbol{r}^{(1)} = Q^{(1)} \cdot \mathrm{vec}(A) + \boldsymbol{g}^{(1)}$ has the same distribution as $Q^{(1)} \cdot \mathrm{vec}(A + \boldsymbol{G})$ by rotational invariance of Gaussian distribution. Now conditioning on $\boldsymbol{r}^{(1)}$, we obtain that $\boldsymbol{r}^{(2)} = Q^{(2)}[\boldsymbol{r}^{(1)}] \cdot \mathrm{vec}(A) + \boldsymbol{g}^{(2)}$ has the *same* distribution as $Q^{(2)}[\boldsymbol{r}^{(1)}] \cdot \mathrm{vec}(A + \boldsymbol{G})$ as $Q^{(2)}[\boldsymbol{r}^{(1)}]$ is orthogonal to $Q^{(1)}$ and hence $Q^{(2)}[\boldsymbol{r}^{(1)}] \cdot \boldsymbol{g}$ is independent of $Q^{(1)}\boldsymbol{g}$, again by rotational invariance.

Thus, we can consider a deterministic algorithm $\mathcal{A}'$ parameterized by the same $\sigma$ which exactly simulates the behavior of $\mathcal{A}$ performing matrix recovery with noisy measurements. Thus,

$$\Pr_{\boldsymbol{G}}[\|\mathcal{A}'(A + \boldsymbol{G}, \sigma) - A\|_\mathsf{F}^2 \le c\|A\|_\mathsf{F}^2] = \Pr_{\boldsymbol{\gamma}}[\|\mathcal{A}(A, \sigma, \boldsymbol{\gamma}) - A\|_\mathsf{F}^2 \le c\|A\|_\mathsf{F}^2].$$

Taking expectation over $\boldsymbol{\sigma}$, we obtain the result. □

Let $\boldsymbol{u}_1, \ldots, \boldsymbol{u}_r$ and $\boldsymbol{v}_1, \ldots, \boldsymbol{v}_r$ be $2r$ random vectors with coordinates being independent standard normal random variables. Let $\boldsymbol{U}$ be an $n \times r$ matrix with columns given by $\boldsymbol{u}_1, \ldots, \boldsymbol{u}_r$ and $\boldsymbol{V}$ be an $n \times r$ matrix with columns given by $\boldsymbol{v}_1, \ldots, \boldsymbol{v}_r$.

**Claim D.2.** If $r \le n/2$, With high probability,

$$\|\sum_{i=1}^r \boldsymbol{u}_i\boldsymbol{v}_i^\mathsf{T}\|_\mathsf{F}^2 = \Theta(n^2 r).$$

*Proof.* Clearly, $\boldsymbol{UV}^\mathsf{T} = \sum_{i=1}^r \boldsymbol{u}_i \boldsymbol{v}_i^\mathsf{T}$. We now have $2nr \geq \|\boldsymbol{V}\|_\mathsf{F}^2 \geq nr/2$ with probability $1 - \exp(-\Theta(nr))$ from [17]. From [26], with probability $\geq 1 - \exp(-\Theta(n))$, $\sigma_{\min}(\boldsymbol{U}) \geq c\sqrt{n}$ for a constant $c$ and as seen in preliminaries, $\|\boldsymbol{U}\|_2 \leq 2\sqrt{n}$. Thus, conditioned on both these events,

$$\|\boldsymbol{UV}^\mathsf{T}\|_\mathsf{F}^2 \geq \sigma_{\min}(\boldsymbol{U})^2 \|\boldsymbol{V}\|_\mathsf{F}^2 \geq c^2 n (nr/2) \geq (c/2)n^2 r$$

and

$$\|\boldsymbol{UV}^\mathsf{T}\|_\mathsf{F}^2 \leq \|\boldsymbol{U}\|_2^2 \|\boldsymbol{V}\|_\mathsf{F}^2 \leq 8n^2 r. \qquad \square$$

Hence $\|(\alpha/\sqrt{n}) \sum_{i=1}^r \boldsymbol{u}_i \boldsymbol{v}_i^\mathsf{T}\|_\mathsf{F}^2 = \Theta(nr)$ with a large probability. Now, we obtain that

$$\Pr_{\boldsymbol{u},\boldsymbol{v},\boldsymbol{G},\boldsymbol{\sigma}}[\|\mathcal{A}'((\alpha/\sqrt{n}) \sum_{i=1}^r \boldsymbol{u}_i \boldsymbol{v}_i^\mathsf{T} + \boldsymbol{G}, \boldsymbol{\sigma}) - (\alpha/\sqrt{n}) \sum_{i=1}^r \boldsymbol{u}_i \boldsymbol{v}_i^\mathsf{T}\|_\mathsf{F}^2 \leq c\|(\alpha/\sqrt{n}) \sum_{i=1}^r \boldsymbol{u}_i \boldsymbol{v}_i^\mathsf{T}\|_\mathsf{F}^2]$$
$$\geq (1 - \exp(-\Theta(n)))p.$$

Thus, there is some $\sigma$ such that

$$\Pr_{\boldsymbol{u},\boldsymbol{v},\boldsymbol{G}}[\|\mathcal{A}'((\alpha/\sqrt{n}) \sum_{i=1}^r \boldsymbol{u}_i \boldsymbol{v}_i^\mathsf{T} + \boldsymbol{G}, \sigma) - (\alpha/\sqrt{n}) \sum_{i=1}^r \boldsymbol{u}_i \boldsymbol{v}_i^\mathsf{T}\|_\mathsf{F}^2 \leq c\|(\alpha/\sqrt{n}) \sum_{i=1}^r \boldsymbol{u}_i \boldsymbol{v}_i^\mathsf{T}\|_\mathsf{F}^2] \geq p(1 - \exp(-\Theta(n))).$$

Thus with probability $\geq p(1 - \exp(-\Theta(n)))$,

$$\|(\sqrt{n}/\alpha)\mathcal{A}'((\alpha/\sqrt{n}) \sum_{i=1}^r \boldsymbol{u}_i \boldsymbol{v}_i^\mathsf{T} + \boldsymbol{G}, \sigma) - \sum_{i=1}^r \boldsymbol{u}_i \boldsymbol{v}_i^\mathsf{T}\|_\mathsf{F}^2 \leq 2cn^2 r.$$

By picking $c$ small enough, we obtain that $t = \Omega(\log(n^2/k)/\log\log n)$ for algorithms with success probability $p \geq 9/10$.

# E    Applications to Other Problems

## E.1    Spectral Low Rank Approximation

**Theorem E.1.** *Given $n, r \in \mathbb{Z}$, if a $t$-round adaptive algorithm that performs $k \geq nr$ general linear measurements in each round is such that for every $n \times n$ matrix $A$, the algorithm outputs a rank $r$ matrix $B$ such that with probability $\geq 99/100$, $\|A - B\|_2 \leq 2\sigma_{r+1}(A)$, then $t \geq c\frac{\log(n^2/k)}{\log\log(n)}$.*

We prove the following helper lemma that we use in our proof.

**Lemma E.2.** *If $A$ is a rank-$r$ matrix and $B$ is an arbitrary matrix satisfying $\|A - B\|_2 \leq t$, then $\|A - [B]_r\|_2 \leq 2t$, where $[B]_r$ denotes the best rank-$r$ approximation of the matrix $B$ in operator norm.*

*Proof.* As $A$ has rank at most $r$, we have $t \geq \|A - B\|_2 \geq \|B - [B]_r\|_2$. Thus, $\|A - [B]_r\|_2 \leq \|A - B\|_2 + \|B - [B]_r\|_2 \leq 2t$. $\square$

*Proof of Theorem E.1.* By a standard reduction, it suffices to show that there is a distribution on $n \times n$ matrices for which any $t$-round *deterministic* algorithm which makes $k$ general linear measurements in each round and outputs a 2-approximate spectral rank approximation satisfies $t \geq c\log(n^2/k)/\log\log(n)$.

Consider the $n \times n$ random matrix $\boldsymbol{M} = \boldsymbol{G} + s\boldsymbol{u}\boldsymbol{v}^\mathsf{T}$ for $s = 500/\sqrt{n}$. Suppose there is a deterministic $t$-round algorithm Alg that makes $k$ linear measurements and outputs rank-$r$ matrix $K = \mathsf{Alg}(\boldsymbol{M})$ such that with probability $\geq 99/100$,

$$\|\boldsymbol{M} - K\|_2 \leq 2\sigma_{r+1}(\boldsymbol{M}) \leq 2\|\boldsymbol{G}\|_2.$$

This implies that $\|s\boldsymbol{u}\boldsymbol{v}^\mathsf{T} - K\|_2 \leq 3\|\boldsymbol{G}\|_2$ and $\|\boldsymbol{u}\boldsymbol{v}^\mathsf{T} - K/s\|_2 \leq 3\|\boldsymbol{G}\|_2/s$.

With probability $\geq 1 - \exp(-\Theta(n))$, we have $2\sqrt{n} \geq \|\boldsymbol{u}\|_2, \|\boldsymbol{v}\|_2 \geq \sqrt{n}/2$ and $\|\boldsymbol{G}\|_2 \leq 3\sqrt{n}$ simultaneously. By a union bound, we have that with probability $\geq 0.98$, $\|\boldsymbol{u}\boldsymbol{v}^\mathsf{T} - K/s\|_2 \leq (9/500)n$.

By Lemma E.1, we have $\|\boldsymbol{u}\boldsymbol{v}^\mathsf{T}-[K/s]_1\|_2 \le (9/250)n$ which implies that $\|\boldsymbol{u}\otimes\boldsymbol{v}-\mathrm{vec}([K/s]_1)\|_2^2 = \|\boldsymbol{u}\boldsymbol{v}^\mathsf{T}-[K/s]_1\|_\mathsf{F}^2 \le 2\|\boldsymbol{u}\boldsymbol{v}^\mathsf{T}-[K/s]_1\|_2^2 \le 2(9/250)^2 n^2$ where we used the fact that the matrix $\boldsymbol{u}\boldsymbol{v}^\mathsf{T}-[K/s]_1$ has rank at most 2. Let $q \in \mathbb{R}^{n^2} = \mathrm{vec}([K/s]_1)/\|[K/s]_1\|_\mathsf{F}$ be a unit vector. We can show that

$$\langle q, \boldsymbol{u}\otimes\boldsymbol{v}\rangle^2 \ge n^2/64$$

by a simple application of the triangle inequality. Thus, using $\mathsf{Alg}$ we can construct a deterministic $t+1$ round algorithm such that the squared projection of $\boldsymbol{u}\otimes\boldsymbol{v}$ onto the query space is at least $\ge n^2/64$ with probability $\ge 98/100$. By (4), we have that $t+1 \ge c'\log(n^2/k)/\log\log(n)$ and therefore we have $t \ge c\log(n^2/k)/\log\log(n)$ for a small enough constant $c$. $\qquad\square$

The following theorem states our lower bound for algorithms which output a spectral LRA for each matrix with high probability.

**Theorem E.3.** *Given $n, r \in \mathbb{Z}$, if a $t$-round adaptive algorithm that performs $k \ge nr$ general linear measurements in each round is such that for every $n \times n$ matrix $A$, the algorithm outputs a rank $r$ matrix $B$ such that with probability $\ge 1-1/\mathrm{poly}(n)$, $\|A-B\|_2 \le 2\sigma_{r+1}(A)$, then $t \ge c\log(n^2/k)$.*

*Proof of Theorem E.3.* The proof of this lemma is similar to that of Theorem E.1. The existence of a randomized $t$ round algorithm that outputs a 2-approximate spectral norm LRA for every instance with probability $\ge 1 - 1/\mathrm{poly}(n)$ implies the existence of a deterministic algorithm that outputs a 2-approximate spectral norm LRA with probability $\ge 1 - 1/\mathrm{poly}(n)$ over the distribution of $\boldsymbol{G} + (\alpha/\sqrt{n})\boldsymbol{u}\boldsymbol{v}^\mathsf{T}$. As in the proof of the Theorem E.1, this algorithm can be used to construct a $t+1$ round algorithm that with probability $\ge 1 - 1/\mathrm{poly}(n)$ over the matrix $\boldsymbol{G} + (\alpha/\sqrt{n})\boldsymbol{u}\boldsymbol{v}^\mathsf{T}$, computes a unit vector $q$ satisfying

$$\langle q, \boldsymbol{u}\otimes\boldsymbol{v}\rangle^2 \ge n^2/64.$$

Now, (5) implies that $t+1 \ge c\log(n^2/k)$ for a small enough constant $c$. $\qquad\square$

For the random matrix $\boldsymbol{M} = \boldsymbol{G} + (\alpha/\sqrt{n})\boldsymbol{u}\boldsymbol{v}^\mathsf{T}$ for a large enough constant $\alpha$ considered in the above theorem, we have that $(\sigma_1(\boldsymbol{M})/\sigma_2(\boldsymbol{M})) \ge 2$ with high probability. Now consider the random matrix

$$\boldsymbol{M}' = \begin{bmatrix} \boldsymbol{M} & 0 \\ 0 & 3\sqrt{n}\alpha I_{r-1} \end{bmatrix}.$$

With high probability, we have $\sigma_r(\boldsymbol{M}') = \sigma_1(\boldsymbol{M}) \ge (\alpha/2)\sqrt{n}$ and $\sigma_{r+1}(\boldsymbol{M}') = \sigma_2(\boldsymbol{M}) \le 2\sqrt{n}$ implying $\sigma_{r+1}(\boldsymbol{M}')/\sigma_r(\boldsymbol{M}') \le 1/2$. A proof similar to that of the above theorem now shows that algorithms using $k \ge nr$ general linear measurements in each round and outputting a 2-approximate rank-$r$ LRA with probability $\ge 1 - 1/\mathrm{poly}(n)$ for matrices $M$ with $\sigma_{r+1}(M)/\sigma_r(M) \le 1/2$ have a lower bound of $\Omega(\log(n^2/k))$ rounds. Moreover for $k \ge Cnr$ for a large enough constant $C$ and $r = O(1)$, the randomized subspace iteration algorithm starting with a subspace of $k/n$-dimensions, after $O(\log(n^2/k))$ rounds outputs, with probability $\ge 1 - 1/\mathrm{poly}(n)$, a 2-approximate rank-$r$ spectral norm LRA for all matrices satisfying $\sigma_{r+1}(M)/\sigma_r(M) \le 1/2$. See [12, Theorem 5.8] for a proof.

Note that the subspace iteration algorithm starting with a subspace of $k/n$-dimensions performs $(k/n) \cdot n = k$ general linear measurements in each round. Thus for $r = O(1)$ and $k \ge Cnr$ for a large enough constant $C$, the lower bound of $\Omega(\log(n^2/k))$ rounds for high probability spectral norm LRA algorithms for "well-conditioned" $(\sigma_{r+1}/\sigma_r \le 1/2)$ instances is tight up to constant factors and shows that general linear measurements offer no improvement over matrix-vector products for well-conditioned problems. Thus we have the following theorem.

**Theorem E.4.** *Any randomized $t$-round algorithm that with probability $\ge 1 - 1/\mathrm{poly}(n)$ outputs a 2-approximate rank-$r$ spectral norm LRA for any arbitrary matrix $A$ satisfying $\sigma_r(A)/\sigma_{r+1}(A) \ge 2$, must have $t = \Omega(\log(n^2/k))$, where $k \ge nr$ is the number of linear measurements the algorithm makes in each round.*

*Moreover, for $r = O(1)$, the subspace iteration algorithm [12] matches the lower bound up to constant factors and outputs a 2-approximate spectral norm LRA for all such instances with probability $\ge 1 - 1/\mathrm{poly}(n)$ in $O(\log(n^2/k))$ rounds.*

## E.2 Symmetric Spectral Norm Low Rank Approximation

An interesting property of the hard distributions from previous works [28, 3] is that those distributions are supported on symmetric matrices and they hence obtain lower bounds for algorithms that work even only on symmetric instances. Although our hard distribution is non-symmetric, we can construct a distribution supported only on symmetric matrices and show lower bounds on algorithms using generalized linear queries for symmetric matrices.

**Theorem E.5.** *Given $n, r \in \mathbb{Z}$, if a $t$-round adaptive algorithm that performs $k \geq nr$ general linear measurements in each round is such that for every $n \times n$ matrix $A$, the algorithm outputs a rank $r \geq 2$ matrix $B$ such that with probability $\geq 99/100$, $\|A - B\|_2 \leq 2\sigma_{r+1}(A)$, then $t \geq c\frac{\log(n^2/k)}{\log \log n}$.*

*Proof.* Consider the $2n \times 2n$ random matrix $M'$ defined as

$$M' = \begin{bmatrix} 0_{n \times n} & M \\ M^\mathsf{T} & 0_{n \times n} \end{bmatrix}$$

where $M = G + suv^\mathsf{T}$ for $s = 500/\sqrt{n}$ and all coordinates of $G, u, v$ are independent standard Gaussian random variables. We have that for $i \in [n]$, $\sigma_{2i-1}(M') = \sigma_{2i}(M') = \sigma_i(M)$ and that any arbitrary $k$ generalized linear measurements of the matrix $M'$ can be simulated using $2k$ general linear measurements of $M$. Now suppose an algorithm that uses $k$ general linear measurements in each round outputs a rank $r \geq 2$ matrix $K$ satisfying

$$\|M' - K\|_2 \leq 2\sigma_{r+1}(M') \leq 2\sigma_1(M).$$

Let the matrix $K$ be of the form

$$K = \begin{bmatrix} K_1 & K_2 \\ K_3 & K_4 \end{bmatrix}$$

As any sub-matrix of $K$ has rank at most that of $K$, we obtain that $K_2$ is a rank-$r$ matrix satisfying

$$\|M - K_2\|_2 \leq 2\sigma_1(M).$$

Thus, the existence of a $t$-round deterministic algorithm that uses $k$ generalized queries in each round and outputs a constant factor approximation of rank $r$, spectral norm LRA, for the random matrix $M'$ with probability $\geq 99/100$ implies the existence of a $t$-round algorithm that uses $2k$ generalized queries in each round and outputs a constant factor approximation of rank-$r$ spectral norm LRA for the random matrix $M$ with probability $\geq 99/100$. Now, as in the proof of Theorem E.1, we obtain that

$$t \geq c\frac{\log(n^2/2k)}{\log \log n} \geq c'\frac{\log(n^2/k)}{\log \log n}.$$

We obtain the proof by appropriately scaling $n$ in the statement. $\square$

The above theorem proves lower bounds for algorithms that solve constant factor rank-$r$ spectral norm Low Rank Approximation (LRA) for all $r \geq 2$ even for symmetric instances. This leaves open just a lower bound on algorithms solving rank 1 spectral norm LRA for symmetric instances.

## E.3 Schatten Norm Low Rank Approximation

We first note the following lemma which bounds the Schatten-$p$ norm of an $n \times n$ Gaussian matrix.

**Lemma E.6** (Equation 3.3 in [19]). *If $G$ is an $n \times n$ matrix with independent entries sampled from $N(0, 1)$ and $p \geq 2$, then with probability $\geq 9/10$, $\|G\|_{\mathsf{S}_p} \leq 30n^{1/2+1/p}$.*

We now state the theorem that shows lower bounds for Schatten norm low rank approximation.

**Theorem E.7.** *Given $n, r \in \mathbb{Z}$, if a $t$-round adaptive algorithm that performs $k \geq nr$ general linear measurements in each round is such that for every $n \times n$ matrix $A$, the algorithm outputs a rank $r$ matrix $B$ such that with probability $\geq 99/100$ we have*

$$\|A - B\|_{\mathsf{S}_p} \leq 2 \min_{\text{rank-}r\ X} \|A - X\|_{\mathsf{S}_p},$$

*then*

$$t \geq c\frac{\log(n^2/k)}{1 + (1/p)\log(n) + \log\log(n)}.$$

*Proof.* The proof is very similar to the proof of Theorem E.1. Let $M = G + suv^\mathsf{T}$ for $s = \alpha/\sqrt{n}$ for $\alpha$ to be chosen later. Let Alg be a $t$-round deterministic algorithm that outputs a 2-approximate Schatten $p$-norm low rank approximation for $M$ with probability $\geq 99/100$ over $M$. Then we have

$$\|uv^\mathsf{T} - K/s\| \leq \frac{3\|G\|_{\mathsf{S}_p}}{s}$$

as $\min_{\text{rank-}r\ X} \|G + suv^\mathsf{T} - X\|_{\mathsf{S}_p} \leq \|G\|_{\mathsf{S}_p}$. Using Lemma E.6, with probability $\geq 9/10$ over $M$, we have that

$$\|\sum_i u_i v_i^\mathsf{T} - K/s\|_{\mathsf{S}_p} \leq 90n^{1+1/p}/\alpha.$$

By picking $\alpha = Bn^{1/p}$ for a large enough constant $B$, we obtain that

$$\|uv^\mathsf{T} - K/s\|_2 \leq \|uv^\mathsf{T} - K/s\|_{\mathsf{S}_p} \leq (9/250)n.$$

Similar to the proof of Theorem E.1, we can construct a unit vector $q \in \mathbb{R}^{n^2}$ such that with probability $\geq 8/10$ over $M$, we have

$$\langle q, u \otimes v\rangle^2 \geq n^2/64.$$

Using (4), we obtain that

$$t + 1 \geq c\frac{\log(n^2/k)}{1 + \log(\alpha) + \log\log(n)} \geq c\frac{\log(n^2/k)}{1 + (1/p)\log(n) + \log\log(n)}.$$

$\square$

In particular, for $p = O(\log(n)/\log\log(n))$, this gives a lower bound of $\Omega(p(2 - \log(k)/\log(n)))$ on the number of adaptive rounds required if an algorithm can query $k$ general linear measurements in each round. Recently, Bakshi, Clarkson and Woodruff [2, Algorithm 2] gave an algorithm for Schatten-$p$ low rank approximation for arbitrary matrices that runs in $O(p^{1/2}\log(n))$[4] iterations to output a constant factor approximation. For instances with $\sigma_r(A)/\sigma_{r+1}(A) = 1 + \Omega(1)$, Saibaba [27, Theorem 8] showed that randomized subspace iteration gives a constant factor approximation to rank-$r$ Schatten-$p$ norm LRA in $O(\log(n))$ iterations using $nr$ linear measurements in each round.

### E.4 Ky-Fan Norm Low Rank Approximation

**Theorem E.8.** *Given $n, r \in \mathbb{Z}$, if a $t$-round adaptive algorithm that performs $k \geq nr$ general linear measurements in each round is such that, for every $n \times n$ matrix A, the algorithm outputs a rank-$r$ matrix B such that with probability $\geq 99/100$ we have*

$$\|A - B\|_{\mathsf{F}_p} \leq 2 \min_{\text{rank-}r\ X} \|A - X\|_{\mathsf{F}_p},$$

*then*

$$t \geq c\frac{\log(n^2/k)}{\log(p) + \log\log(n)}.$$

*Proof.* Again consider the same instance $M = G + suv^\mathsf{T}$ with $s = \alpha/\sqrt{n}$ for $\alpha$ chosen later. If $K$ is a 2-approximate rank-$r$ low rank approximation of $M$ in Ky-Fan $p$ norm, then

$$\|G + suv^\mathsf{T} - K\|_{\mathsf{F}_p} \leq 2\|G\|_{\mathsf{F}_p}$$

which by the triangle inequality implies that $\|s\sum_{i=1}^r u_i v_i^\mathsf{T} - K\|_{\mathsf{F}_p} \leq 3\|G\|_{\mathsf{F}_p}$. As $\|G\|_2 \leq 2\sqrt{n}$ with high probability, we have that $\|G\|_{\mathsf{F}_p} \leq 2p\sqrt{n}$ with high probability. Thus,

$$\|s\sum_{i=1}^r u_i v_i^\mathsf{T} - K/s\|_2 \leq \|s\sum_{i=1}^r u_i v_i^\mathsf{T} - K/s\|_{\mathsf{F}_p} \leq \frac{6pn}{\alpha}.$$

---

[4]Although their algorithm is stated to use $rp^{1/6}\log^2(n)$ matrix-vector products, this does not appear to have been optimized, and looking at the analysis it runs in at most $p^{1/2}\log(n)$ iterations, where each round queries at most $r$ matrix-vector products.

Picking $\alpha = Bp$ for a large enough constant $B$, we obtain that using the matrix $K$, we can construct a unit vector $q$ such that $\langle q, \boldsymbol{u} \otimes \boldsymbol{v} \rangle^2 \geq n^2/64$, thus showing a

$$t = c\frac{\log(n^2/k)}{\log(p) + \log\log(n)}$$

round lower bound on any algorithm that performs $k$ general linear measurements to output a 2-approximate rank-$r$ approximation in Ky-Fan $p$ norm. □

For $p = O(1)$, the Block Krylov iteration algorithm [20] gives an $O(1)$ approximate solution to the rank-$r$ Ky-Fan norm low rank approximation problem in $O(\log(n))$ rounds while querying $nr$ general linear measurements in each round. Thus our lower bound on constant approximate algorithms for Ky-Fan norm LRA is optimal up to a $\log\log(n)$ factor for $r, p = O(1)$.

### E.5  Frobenius Norm Low Rank Approximation

**Theorem E.9.** *Given an $n \times n$ matrix $A$ with $\sigma_1(A)/\sigma_2(A) \geq 2$, if a $t$-round algorithm outputs a rank-1 matrix $B$ satisfying $\|A - B\|_{\mathsf{F}}^2 \leq (1 + 1/n)\|A - [A]_1\|_{\mathsf{F}}^2$ with probability $\geq 99/100$, then we have $t \geq c\log(n^2/k)/\log\log(n)$.*

First, we prove the following helper lemma.

**Lemma E.10.** *Given $A \in \mathbb{R}^{n \times n}$, if a rank $r$ matrix $B$ is a $1 + 1/(n - r)$ approximate rank $r$ Frobenius norm LRA, then*

$$\|A - B\|_2^2 \leq 2\sigma_{r+1}(A)^2.$$

*Proof.* We use the fact [12, Theoem 3.4] that if a rank $r$ matrix satisfies $\|A - B\|_{\mathsf{F}}^2 \leq \|A - [A]_r\|_{\mathsf{F}}^2 + \eta$, then $\|A - B\|_2^2 \leq \|A - [A]_r\|_2^2 + \eta$. If $B$ is a $1 + 1/(n - r)$ approximate solution for rank $r$ Frobenius norm LRA, we have

$$\|A - B\|_{\mathsf{F}}^2 \leq \left(1 + \frac{1}{n - r}\right)\|A - [A]_r\|_{\mathsf{F}}^2$$

$$\leq \|A - [A]_r\|_{\mathsf{F}}^2 + \frac{\sigma_{r+1}(A)^2 + \ldots + \sigma_n(A)^2}{n - r}$$

$$\leq \|A - [A]_r\|_{\mathsf{F}}^2 + \sigma_{r+1}(A)^2$$

which implies $\|A - B\|_2^2 \leq \|A - [A]_r\|_2^2 + \sigma_{r+1}(A)^2 = 2\sigma_{r+1}(A)^2$. □

*Proof of Theorem E.9.* Using the above lemma, we have that whenever $\|A - B\|_{\mathsf{F}}^2 \leq (1 + 1/n)\|A - [A]_1\|_{\mathsf{F}}^2$ for a rank 1 matrix $B$, then $\|A - B\|_2^2 \leq 2\|A - B\|_2^2$. Consider the random matrix $\boldsymbol{M} = \boldsymbol{G} + (\alpha/\sqrt{n})\boldsymbol{u}\boldsymbol{v}^\mathsf{T}$ considered in Theorem E.1. With probability $\geq 99/100$, we have that $\sigma_1(\boldsymbol{M})/\sigma_2(\boldsymbol{M}) \geq 2$ by picking $\alpha$ to be a large enough constant.

A $t$-round randomized algorithm for $(1 + 1/n)$-approximate rank-1 Frobenius norm LRA thus implies the existence of a $(t + 1)$ round deterministic algorithm that with probability $\geq 98/100$ over the random matrix $\boldsymbol{G} + (\alpha/\sqrt{n})\boldsymbol{u}\boldsymbol{v}^\mathsf{T}$ makes general queries such that the squared projection of $\boldsymbol{u} \otimes \boldsymbol{v}$ onto the query space of the algorithm exceeds $n^2/64$ using similar arguments as in proof of Theorem E.1. Now, using (4), we obtain that

$$t \geq c\log(n^2/k)/\log\log(n)$$

for a small enough constant $c$. □

For matrices $A$ with $\sigma_1(A)/\sigma_2(A) \geq 2$, subspace iteration algorithm gives a $1 + 1/n$-approximate solution in $O(\log(n))$ rounds using $n$-linear measurements [10, 20]. The above lower bound shows that if an algorithm is allowed $o(\log(n)/\log\log(n))$ adaptive rounds, then the algorithm has to make $n^{2-o(1)}$ linear measurements in each round, which is almost as many measurements as is required to read the entire matrix.

## E.6 Lower Bound for Reduced-Rank Regression in Spectral Norm

Given matrices $A$, $B$, and a rank parameter $r$, the reduced-rank regression problem in spectral norm is defined as

$$\min_{\text{rank-}r\ X} \|AX - B\|_2.$$

Taking $A = I$, we have that the spectral norm low rank approximation is a special case of the reduced-rank regression problem. Thus we have the following lower bound on the number of rounds of an adaptive algorithm that outputs a 2-approximate solution for the reduced-rank regression problem.

**Theorem E.11.** *If a $t$-round adaptive algorithm which given arbitrary $n \times n$ matrices $A, B$ and a parameter $r$ outputs a 2-approximate solution for the reduced-rank regression problem with probability $\geq 9/10$, then $t = \Omega(\log(n^2/k)/\log\log(n))$ where $k$ is the number of linear measurements of the matrices $A$ and $B$ that the adaptive algorithm performs in each of the $t$ rounds.*

With $nr$ adaptive linear measurements in each round, the above theorem gives a lower bound of $\Omega(\log(n/r)/\log\log(n))$ on the number of rounds required to obtain a factor 2 approximation. Kacham and Woodruff [15] give an algorithm for reduced-rank regression that outputs constant factor approximations to the reduced-rank problem in $O(\text{poly}(\log n))$ (ignoring polylogarithmic factors from condition numbers) adaptive rounds using $nr$ linear measurements in each round.

If the matrix $B = M + P$ has the structure of a planted matrix studied in this paper with $P$ being a rank $r$ matrix and $\|P\|_2 > 10\|B - [B]_r\|_2$ and if the matrix $A$ is well conditioned, then we can find an $O(1)$ approximate solution to reduced-rank regression problem in $O(\log(n) + \log(\|P\|_2/\text{OPT}))$ rounds, where OPT is the optimal value of the reduced-rank regression problem.

First, we find a rank $r$ matrix $P'$ such that $\|B - P'\|_2 \leq 2\|B - [B]_r\|_2$ in $O(\log(n))$ adaptive rounds using the Block Krylov iteration algorithm of [20], which queries $nr$ linear measurements in each round. Now, for any matrix $X$,

$$\|AX - P'\|_2 = \|AX - B\|_2 \pm \|B - P'\|_2 = \|AX - B\|_2 \pm 2 \cdot \text{OPT}.$$

Let $P' = U\Sigma V^\mathsf{T}$ be the singular value decomposition with matrix $U\Sigma$ having $r$ columns. Then using high precision regression routines (see [33] and references therein), we find a matrix $\tilde{X}$ such that

$$\|A\tilde{X} - AA^+ U\Sigma\|_\mathsf{F}^2 \leq \varepsilon \|U\Sigma\|_\mathsf{F}^2 \leq 10 r \varepsilon \|P\|_2^2.$$

in $O(\log(1/\varepsilon))$ iterations, where each iteration queries $nr$ linear measurements. Again, using the triangle inequality, we have

$$\|A\tilde{X}V^\mathsf{T} - P'\|_2 = \|A\tilde{X} - U\Sigma\|_2 \leq \|AA^+ U\Sigma - A\tilde{X}\|_2 + \|AA^+ U\Sigma - U\Sigma\|_2.$$

If $X^*$ is the optimal solution for the reduced rank regression problem, we have

$$
\begin{aligned}
\|AA^+ U\Sigma - U\Sigma\|_2 &= \|AA^+ U\Sigma V^\mathsf{T} - U\Sigma V^\mathsf{T}\|_2 \\
&\leq \|AX^* - P'\|_2 \\
&\leq \|AX^* - B\|_2 + 2 \cdot \text{OPT} = 3 \cdot \text{OPT}.
\end{aligned}
$$

We also have $\|AA^+ U\Sigma - A\tilde{X}\|_2 \leq \|AA^+ U\Sigma - A\tilde{X}\|_\mathsf{F} \leq O(\sqrt{r\varepsilon}\|P\|_2)$. We set $\varepsilon = (\text{OPT}/\|P\|_2)^2/Kr$ for a large enough constant $K$ and obtain that $\|AA^+ U\Sigma - A\tilde{X}\|_2 \leq \text{OPT}$. Overall, we have $\|A\tilde{X}V^\mathsf{T} - B\|_2 \leq \|A\tilde{X}V^\mathsf{T} - P'\|_2 + 2 \cdot \text{OPT} \leq 6 \cdot \text{OPT}$. Thus we obtain a 6-approximate solution in $O(\log(n) + \log(\|P\|_2/\text{OPT}))$ iterations given that $A$ is well-conditioned and $B$ is of the form $M + P$ with $\|P\|_2 \geq 10\|M\|_2$ for a rank $r$ matrix $P$. Thus the lower bound is near optimal for algorithms that output $O(1)$ approximate solution for planted models and well-conditioned coefficient matrices.

## E.7 Lower Bound for Approximating the $i$-th Singular Vectors

**Theorem E.12.** *If a $t$-round algorithm, given an $n \times n$ matrix $A$ and $i \in [n]$, outputs unit vectors $u_i'$ and $v_i'$ such that with probability $\geq 9/10$,*

$$\|u_i' - u_i\|_2 \leq 1/10 \text{ and } \|v_i' - v_i\|_2 \leq 1/10$$

*where $u_i$ and $v_i$ are respectively the $i$-th left and right singular vectors of the matrix $A$, then $t = \Omega(\log(n^2/k)/\log\log(n))$, where $k$ is the number of linear measurements the algorithm performs on the matrix $A$ in each of the $t$ rounds.*

*We also have a $t = \Omega(\log(n^2/k))$ lower bound on the number of rounds required for an adaptive algorithm to compute approximations to the $i$-th left and right singular vectors with probability $\geq 1 - 1/\operatorname{poly}(n)$.*

*Proof.* Consider the $n \times n$ random matrix $\boldsymbol{M} = \boldsymbol{G} + (\alpha/\sqrt{n})\boldsymbol{u}\boldsymbol{v}^\mathsf{T}$ for a large enough constant $\alpha$. The existence of a $t$-round algorithm as in the statement implies that there is a deterministic $t$-round algorithm that outputs approximations to singular vectors of the matrix $\boldsymbol{M}$ with probability $\geq 9/10$. We then have that $10\|\boldsymbol{G}\|_2 \leq \|(\alpha/\sqrt{n})\boldsymbol{u}\boldsymbol{v}^\mathsf{T}\|_2$ and $\|\boldsymbol{u}\|_2, \|\boldsymbol{v}\|_2 \geq \sqrt{n}/2$ with large probability. Condition on this event. Let $v_1 \in \mathbb{R}^n$ be the top right singular vector of the matrix $\boldsymbol{M}$. We have

$$(9/10)\|(\alpha/\sqrt{n})\boldsymbol{u}\boldsymbol{v}^\mathsf{T}\|_2 \leq \|(\alpha/\sqrt{n})\boldsymbol{u}\boldsymbol{v}^\mathsf{T}\|_2 - \|\boldsymbol{G}\|_2 \leq \|\boldsymbol{M}\|_2 = \|(\boldsymbol{G} + (\alpha/\sqrt{n})\boldsymbol{u}\boldsymbol{v}^\mathsf{T})v_1\|_2.$$

By the triangle inequality, we have

$$\begin{aligned}
\|(\alpha/\sqrt{n})\boldsymbol{u}(\boldsymbol{v}^\mathsf{T}v_1)\|_2 &\geq \|(\boldsymbol{G} + (\alpha/\sqrt{n})\boldsymbol{u}\boldsymbol{v}^\mathsf{T})\|_2 - \|\boldsymbol{G}v_1\|_2 \\
&\geq (9/10)\|(\alpha/\sqrt{n})\boldsymbol{u}\boldsymbol{v}^\mathsf{T}\|_2 - (1/10)\|(\alpha/\sqrt{n})\boldsymbol{u}\boldsymbol{v}^\mathsf{T}\|_2 \\
&= (8/10) \cdot (\alpha/\sqrt{n})\|\boldsymbol{u}\|_2\|\boldsymbol{v}^\mathsf{T}\|_2.
\end{aligned}$$

Thus, we obtain that $|\boldsymbol{v}^\mathsf{T}v_1| \geq 4/5\|\boldsymbol{v}\|_2$. Similarly, if $u_1$ is the top left singular vector, we have that $|\boldsymbol{u}^\mathsf{T}u_1| \geq 4/5\|\boldsymbol{u}\|_2$. Suppose $v_1'$ is a unit vector such that $\|v_1 - v_1'\|_2 \leq 1/10$. Then, $|\boldsymbol{v}^\mathsf{T}v_1'| \geq |\boldsymbol{v}^\mathsf{T}v_1| - (1/10)\|\boldsymbol{v}\|_2 \geq (7/10)\|\boldsymbol{v}\|_2$. Similarly, $|\boldsymbol{u}^\mathsf{T}u_1'| \geq (7/10)$ which implies

$$\langle u_1' \otimes v_1', \boldsymbol{u} \otimes \boldsymbol{v}\rangle^2 = \langle u_1', \boldsymbol{u}\rangle^2 \langle v_1', \boldsymbol{v}\rangle^2 \geq (49/100)\|\boldsymbol{u}\|_2^2\|\boldsymbol{v}\|_2^2 \geq n^2/10.$$

Thus, with a large probability over the random matrix $\boldsymbol{M}$, approximations to the top left and right singular vectors of the matrix $\boldsymbol{M}$ can be used to construct an $n^2$-dimensional unit vector $q$ such that $\langle q, \boldsymbol{u} \otimes \boldsymbol{v}\rangle^2 \geq n^2/10$. Thus, we have a $t + 1$ round deterministic algorithm such that the squared projection of $\boldsymbol{u} \otimes \boldsymbol{v}$ onto the query space of the algorithm is $\geq n^2/10$ with probability $\geq 8/10$ over the random matrix $\boldsymbol{M}$. Now, (4) implies that $t + 1 \geq c\log(n^2/k)/\log\log(n)$ for a small enough constant $c$.

To extend the above lower bound to approximating $i$-th singular vector for all $i \leq n/2$, we can use the following instance:

$$\boldsymbol{M}' = \begin{bmatrix} \boldsymbol{M} & 0 \\ 0 & S \end{bmatrix}$$

where $\boldsymbol{M}$ is the $(n/2) \times (n/2)$ random matrix $\boldsymbol{G} + (\alpha/\sqrt{n/2})\boldsymbol{u}\boldsymbol{v}^\mathsf{T}$ and $S$ is a deterministic diagonal matrix chosen such that the top singular vectors of $\boldsymbol{M}$ correspond to $i$-th singular vectors of the matrix $\boldsymbol{M}'$. If $S$ is chosen to be the diagonal matrix with $i - 1$ values equal to $10\sqrt{n}\alpha$ and the rest of the entries are set to 0, we have that with high probability that the $i$-th left and right singular vectors of $\boldsymbol{M}'$ correspond to the top left and right singular vectors of $\boldsymbol{M}$ as $\|M\|_2 \leq 10\sqrt{n}\alpha$ with high probability. Now the existence of a $t$-round randomized algorithm to approximate $i$-th left and right singular vectors of an $n \times n$ matrix for $i \leq n/2$ with probability $\geq 9/10$ implies that there is a deterministic $t$-round algorithm that approximates the top left and right singular vectors of the $(n/2) \times (n/2)$ matrix $\boldsymbol{M}$ with probability $\geq 8/10$. From the above argument we have $t + 1 = \Omega(\log((n/2)^2/k)/\log\log(n/2)) = \Omega(\log(n^2/k)/\log\log(n))$.

For $i \geq n/2$, we can do a similar reduction. We have that the top left singular vector $u_1$ of the $(n/2) \times (n/2)$ matrix $\boldsymbol{M}$ is same as the top singular (eigen-) vector of the matrix $\boldsymbol{M}\boldsymbol{M}^\mathsf{T}$ and with high probability is equal to the last singular (eigen-) vector of the matrix $(100n\alpha^2)I - \boldsymbol{M}\boldsymbol{M}^\mathsf{T}$. Using the same trick as above, we can create an $n \times n$ matrix such that the $i$-th singular vector for $i \geq n/2$ corresponds to $u_1$. Instead of $\boldsymbol{M}\boldsymbol{M}^\mathsf{T}$, if we use the matrix $\boldsymbol{M}^\mathsf{T}\boldsymbol{M}$, then we can make the top right singular vector $v_1$ of the matrix $\boldsymbol{M}$ correspond to the $i$-th singular vector of an $n \times n$ matrix. Thus, for all $i \in [n]$, we have an $\Omega(\log(n^2/k)/\log\log(n))$ lower bound for any randomized algorithm that computes the $i$-th left and right singular vectors of an arbitrary $n \times n$ matrix $A$ using $k$ linear measurements of $A$ in each round of the algorithm.

The lower bounds for algorithms that output approximations to singular vectors with probability $\geq 1 - 1/\operatorname{poly}(n)$ follow similarly using the high probability lower bounds from (5). $\qquad\square$

### E.8 Lower Bounds for Sparse Matrices

The following theorem gives a lower bound of $\Omega(\log(m/k))$ rounds on adaptive algorithms that compute 2-approximate spectral norm LRA for matrices with at most $m$ nonzero entries.

**Theorem E.13.** *Any $t$ round randomized algorithm that computes, given an arbitrary $n \times n$ matrix $A$ with at most $m$ nonzero entries, a 2-approximate rank $r$ spectral norm LRA of $A$ with probability $\geq 1 - 1/\operatorname{poly}(m)$, must have $t = \Omega(\log(m/k))$, where $k$ is the number of general linear measurements queried by the algorithm in each of the $t$ rounds.*

*Proof.* We consider the distribution of the $\sqrt{m} \times \sqrt{m}$ random matrix $\boldsymbol{G} + (\alpha/m^{1/4})\boldsymbol{u}\boldsymbol{v}^\mathsf{T}$ for a large enough constant $\alpha$ and embed this instance into an $n \times n$ matrix. Clearly, all the matrices drawn from this distribution have at most $m$ nonzero entries. And from Theorem E.3, any deterministic algorithm using $k$ linear measurements in each round and computing a 2-approximate spectral norm approximation for a matrix drawn from this distribution, with probability $\geq 1 - 1/\operatorname{poly}(m)$, must use $\Omega(\log((\sqrt{m})^2/k)) = \Omega(\log(m/k))$ rounds.

Thus by Yao's minimax lemma, any randomized algorithm that outputs a 2-approximate spectral norm approximation with probability $\geq 1 - 1/\operatorname{poly}(m)$ for an arbitrary matrix with $m$ nonzero entries must use $t = \Omega(\log(m/k))$ rounds. $\qquad\square$

The above theorem implies that any algorithm with only $O(1)$ adaptive rounds must query $\Omega(m)$ linear measurements. This lower bound is tight up to constant factors as with $2m$ linear measurements, since all of the $m$ non-zero entries of any arbitrary matrix can be computed in just 1 round using a Vandermonde matrix [8, Theorem 2.14].

