# OpenReview forum: "Lower Bounds on Adaptive Sensing for Matrix Recovery"
_NeurIPS.cc/2023/Conference — NeurIPS 2023 poster_

### Official Review · Reviewer_Txfq · 2023-06-27

**Soundness:** 3 good
**Presentation:** 3 good
**Contribution:** 3 good
**Rating:** 7
**Confidence:** 3

**Summary:**

Sparse recovery for vectors has been studied for a long time. It has been shown that allowing adaptive queries will give extra power to reduce the number of queries. We also have upper and lower bounds in this setting. More recently, sparse recovery for low-rank matrices has also been extensively studied. With non-adaptive queries, $\Omega(n^2)$ queries are necessary. With adaptive queries, based on the power method, we get an algorithm with $O(nr)$ linear measurements in each round over $O(\log n)$ adaptive rounds. It is an interesting question to see if the power method is optimal in this setting.

This paper gives an affirmative answer to this question by showing a measurement-vs-rounds trade-off for recovering low-rank matrices using linear measurements. More specifically, let $A$ be an $n$-by-$n$ matrix with rank $r$. In each round, the algorithm can make $k$ queries. Each query is an $n$-by-$n$ matrix $S$, and the answer is $\langle S, A\rangle$ with some Gaussian noise. The goal is to reconstruct a matrix $\hat{A}$ such that $\|\hat{A}-A\|_F\leq c\|A\|_F$. The main result of this paper is that any adaptive algorithm which uses $n^{2-\beta}$ linear measurements in each round must run for $o(\log n/ \log \log n)$ rounds to compute a good reconstruction with high probability. Their techniques also apply to obtain measurements-vs-rounds trade-offs for other numerical linear algebra problems, including low-rank approximation in several different norms, singular vector approximation, etc.

Technically, the construction of the hard instance is $A=\frac{\alpha}{\sqrt{n}}\sum_{i=1}^r u_iv_i^\top$, where $u_i,v_i$ are independent, Gaussian random vectors. In the proof of their main result, they first reduce to show the lower bound for a deterministic algorithm with perfect linear measurements of the random matrix $\frac{\alpha}{\sqrt{n}}\sum_{i=1}^r u_iv_i^\top+G$, where $G$ is a Gaussian matrix, outputting a reconstruction of $\frac{\alpha}{\sqrt{n}}\sum_{i=1}^r u_iv_i^\top$. The key observation is that the distribution of the responses in the first round is close to $N(0, I_k)$, and therefore the algorithm cannot have a lot of “information” about the target matrix. The proof relies on a random tensor concentration result and Bayes risk lower bounds.


**Strengths:**

This paper fills a gap in the field of sparse recovery and makes significant progress in understanding the limitations of adaptive queries in solving numerical linear algebra problems. Their main result is the lower bound for the sparse recovery of low-rank matrices, nearly matching the upper bound via the power method. Even if they use some techniques in prior works, the proofs are still non-trivial. This paper is well-motivated, and the idea of their techniques and proofs are clearly presented. Most of the claims are sound to me.

**Weaknesses:**

For the applications, it is difficult to judge the significance since there is not enough comparison between the results in this paper and prior works. It seems that some applications are for the regimes incomparable to the previous literature.

**Questions:**

1. Does the lower bound require that all the iterations take $k$ queries uniformly? What if the algorithm can adaptively decide the number of queries in each iteration?

2. It would be better to show some known upper bounds for the problems in Table 1.

3. Line 52: $M\in \mathbb{R}^{n\times n} \rightarrow \mathbb{R}^t$. $t$ should be $k$.


**Limitations:**

N/A.

---

> ### Author Rebuttal · Authors · 2023-08-09
>
> - Uniform number of measurements: We note that our lower bound says that even when one is allowed $n^{2 - \beta}$ linear measurements in *each* round, the algorithm must use $o(\log n/ \log \log n)$ rounds to be able to approximate the matrix. So, allowing for the number of measurements to be adaptively chosen in each round does not seem to modify the results.
> - Known Upper bounds: In the appendix, we show that under certain conditions, the 2-approximate spectral norm approximation problem can be solved in $O(\log n^2/k)$ rounds using $k$ linear measurements in each round using the subspace iteration algorithm. We will include a discussion in the paper on the upper bounds for other problems and add references to them.

---

> > ### Comment · Reviewer_Txfq · 2023-08-14
> >
> > I thank the authors for their response. I keep my score.

---

### Official Review · Reviewer_cbbb · 2023-07-06

**Soundness:** 4 excellent
**Presentation:** 4 excellent
**Contribution:** 2 fair
**Rating:** 5
**Confidence:** 2

**Summary:**

This paper focuses on investigating the lower bound of the adaptive low-rank matrix sensing problem. Specifically, the authors demonstrate that when the noise level significantly exceeds the signal, any adaptive algorithm requiring fewer than $o(\log(n)/\log\log n)$ rounds must utilize at least $\Omega(n^2)$ linear measurements in total. This finding highlights an intriguing trade-off between the number of measurements and the number of rounds in various matrix sensing problems within numerical linear algebra.

The paper presents a clear message, and the theoretical results are robust and captivating. However, one aspect worth considering is the general interest in the noise level examined in this study.

**Strengths:**

- The theoretical results are very sound
- The presentation is very clean and easy to follow
- The adaptive settings of matrix recovery is of general interest

**Weaknesses:**

This paper examines the case where the noise level is assumed to be $O(1)$, while the signal of each entry is considered to be $O(1/\sqrt{N})$. It is worth noting that in a more typical scenario where both signals and noises are of magnitude $O(1)$, a single round with $\Omega(nr)$ measurements, as presented in [5], would suffice. This raises questions about the significance of studying this extreme case and the role that $\sigma_l$ plays in the lower-bound tradeoff.

To provide more justifications for studying this extreme case, it would be beneficial for the authors to elaborate on the motivations and implications behind their choice of noise and signal levels. By offering insights into why this specific scenario is relevant and shedding light on the insights gained from this extreme case analysis, the authors can strengthen the value and message of their paper.

**Questions:**

The authors make an assumption that $M^{(i)}$ represents an orthonormal basis and argue that this assumption does not affect the generality of their approach due to the possibility of a change of basis. However, it is important to consider whether this change of basis has any impact on the homogeneous noise level, i.e., $Ag$ for a Gaussian vector $g$ is not a homogeneous Gaussian vector anymore if $A$ is not unitary. It would be valuable for the authors to provide further elaboration on this point.

**Limitations:**

I do not forsee any potential negative societal impact of this work.

---

> ### Author Rebuttal · Authors · 2023-08-09
>
> - Noise level: You are correct that the lower bounds we study are in the high-noise regime. A main reason for this is that adaptivity does not reduce the number of linear measurements by a large factor in the low noise regime since the non-adaptive sensing algorithms already recover the low rank matrix using a number of linear measurements of approximately $O(nr)$, up to some small multiplicative factors. So, it does seem that  adaptive algorithms are majorly helpful only in the high noise regime, which is where the study of measurements vs rounds tradeoff becomes interesting. While not exactly related to matrix recovery, [1] studies the performance of data-aware projection algorithms when the matrix is corrupted with a large noise.
> - Orthonormality of the measurements: We will elaborate on this. You are correct that the assumption about orthonormal measurements is not without loss of generality. For example, the standard assumptions in matrix recovery allow the same linear measurement to be performed multiple times and obtain results with independent noise in each of the measurements whereas our lower bounds which assume that the measurements are orthonormal does not account for this scenario. We will emphasize this in the next version. We do note that this loss of generality is not an issue for the lower bounds that we present for other problems such as spectral norm low rank approximation, etc.
>
> [1] Abdullah, Amirali, et al. "Spectral approaches to nearest neighbor search." 2014 IEEE 55th Annual Symposium on Foundations of Computer Science. IEEE, 2014.

---

### Official Review · Reviewer_m7zx · 2023-07-06

**Soundness:** 2 fair
**Presentation:** 2 fair
**Contribution:** 3 good
**Rating:** 6
**Confidence:** 2

**Summary:**


In this paper, the authors discuss the power of adaptive algorithms in low-rank approximation. This is a setting where one observes general linear measurements of a matrix and wants to produce the best r-rank approximation of it.

It is known that non-adaptive algorithms need order n^2 measurements and that with log n rounds a spectral algo works with nearly linear measurements.

The authors discuss whether something can be done with o(log n) rounds.
## Contribution

The authors prove that for an algo that works with o(log n/log log n) rounds, we need order n^{2-o(1)} measurements.




**Strengths:**

The authors point to the interesting importance of having access to approx log n rounds, as they prove that having access to o(log n/log log n) rounds is like having one round in terms of measurement complexity. I find the contribution clean, correct, and interesting.

**Weaknesses:**

It is a little bit disappointing that the authors cannot only assume o(log n) rounds, and need to assume o(log n/log log n) rounds. Could they comment more on that weakness?



**Questions:**

It seems that for the lower bound the authors only assume that the target matrix is r-rank Gaussian spike plus Gaussian noise. Is that true?

If so, please highlight it as this is a rather simple "bad" example to build a lower bound, which makes the result even more appealing.

**Limitations:**

See above.

---

> ### Author Rebuttal · Authors · 2023-08-09
>
> - $o(\log n)$ vs $o(\log n / \log\log n)$: Our Bayes risk analysis, which union bounds over the information growth in each of the rounds, is what leads to the $\Omega(\log n/\log \log n)$ rounds lower bounds instead of the more desirable $\Omega(\log n)$ rounds lower bound. However, we do note that, as stated in Eq (5), if we want algorithms that succeed with probability $\ge 1 - 1/\text{poly}(n)$, we can obtain an $\Omega(\log n)$ adaptive round lower bound on any algorithm that uses $n^{2 - \beta}$ linear measurements in each round.
> - Hard instance being rank-$r$ gaussian + Gaussian: Yes, the distribution of matrices for which we prove the lower bound is a rank-$r$ Gaussian (sum of $r$ outer products of independent Gaussian vectors) + another Gaussian. We will highlight this.

---

> > ### Comment · Reviewer_m7zx · 2023-08-20
> >
> > I thank the authors for their response and I maintain my score.

---

### Official Review · Reviewer_NmzP · 2023-07-07

**Soundness:** 4 excellent
**Presentation:** 3 good
**Contribution:** 3 good
**Rating:** 7
**Confidence:** 2

**Summary:**

This paper studies the problem of low-rank matrix reconstruction from linear measurements, which is a matrix generalization of the well known sparse reconstruction setting. The provide new lower bounds for this problem under different error metrics, such as the Frobenius norm, essentially showing that non-trivial reconstruction error is impossible unless $\Omega(\log n)$ adaptive rounds of measurements are made. The hard instance is a rank-1 matrix planted into i.i.d Gaussian noise.

**Strengths:**

- The problem solved is fundamental and to the best of my knowledge the contributions are new and generally applicable.
- The technical steps are clearly explained and well written, and look sound.

**Weaknesses:**

- The presentation in Section 1 and 1.3 could be improved. While I like that the explanation goes into technical detail, it could be significantly improved by having an outline of the 2-3 most significant contributions and a more modular presentation.

**Questions:**

Is there any potential connection between this work and the hardness results used in [1] for matrix completion based on planted clique (or the references within)? Are there any implications for matrix completion?

[1] Yudong Chen, Incoherence-Optimal Matrix Completion, IEEE Transactions on Information Theory, 2015

---

> ### Author Rebuttal · Authors · 2023-08-09
>
> - Thanks for the comments and question. Since the value of an entry of a matrix can be computed using a linear measurement, it is indeed true that lower bounds on linear measurements must also carry over to the matrix completion setting. As our lower bounds are in the “noisy” linear measurements model, they are not quite applicable to your reference that studies the recovery of exact matrices using the entries that are randomly observed. When there is no noise, an $n \times n$ rank-$r$ matrix can be recovered using only $O(nr)$ linear measurements over two adaptive rounds as follows: In the first round, compute $AS$ where $S$ is an $n \times O(r)$ matrix with independent Gaussian entries. Note that $AS$ can be computed using $O(nr)$ linear measurements. We can show that with a large probability $\text{rank}(AS) = \text{rank}(A)$ and hence the column space of $AS$ is the same as the column space of $A$. If then $U$ is an orthonormal basis for the column space of $A$, we can compute $U^T A$ using another $O(nr)$ linear measurements. So in the noiseless setting, we can recover the rank $r$ matrix using only $O(nr)$ linear measurements over 2 rounds as opposed to lower bounds in our setting that says one needs ~ $O(n^2)$ measurements if we want to recover the matrix in $o(\log n/ \log \log n)$ rounds.
>
> - In the relevant setting of matrix recovery with noise (e.g., [1]), the previous works seem to study only very low noise regimes. So the tightness of the results cannot be inferred from our lower bounds.
>
> [1] Candes, Emmanuel J., and Yaniv Plan. "Matrix completion with noise." Proceedings of the IEEE 98.6 (2010): 925-936.

---

> > ### Comment · Reviewer_NmzP · 2023-08-18
> >
> > I thank the authors for the response.

---

### Decision · Program_Chairs · 2023-09-21

**Decision:**

Accept (poster)

**Comment:**

In this paper, the authors investigated fundamental lower bounds for adaptive sensing in noisy low-rank matrix recovery problems, focusing on the interplay between adaptivity and sample complexity.  When the signal-to-noise is approaching the information-theoretic limit, it was shown previously that non-adaptive sensing algorithms require n^2 measurements, which can be improved with the aid of log n rounds of adaptive sensing. It is then natural to ask whether log n rounds of adaptive sensing is necessary. The authors tackled this issue by developing a lower bound showing that: any algorithm that uses no more than o(log n / loglog n) rounds of adaptive sensing necessarily requires at least n^{2-o(1)} measurements, thus justifying the provable benefits of adaptive sensing. It would be nice if the authors could further extend their theory to accommodate o(log n) rounds of adaptivity, so as to fully close the gap.